# DIRECTIONAL GRAPH NETWORKS

## ABSTRACT

In order to overcome the expressive limitations of graph neural networks (GNNs), we propose the first method that exploits vector flows over graphs to develop globally consistent directional and asymmetric aggregation functions. We show that our directional graph networks (DGNs) generalize convolutional neural networks (CNNs) when applied on a grid. Whereas recent theoretical works focus on understanding local neighbourhoods, local structures and local isomorphism with no global information flow, our novel theoretical framework allows directional convolutional kernels in any graph. First, by defining a vector field in the graph, we develop a method of applying directional derivatives and smoothing by projecting node-specific messages into the field. Then we propose the use of the Laplacian eigenvectors as such vector field, and we show that the method generalizes CNNs on an $n$-dimensional grid, and is provably more discriminative than standard GNNs regarding the Weisfeiler-Lehman 1-WL test. Finally, we bring the power of CNN data augmentation to graphs by providing a means of doing reflection, rotation and distortion on the underlying directional field. We evaluate our method on different standard benchmarks and see a relative error reduction of 8% on the CIFAR10 graph dataset and 11% to 32% on the molecular ZINC dataset. An important outcome of this work is that it enables to translate any physical or biological problems with intrinsic directional axes into a graph network formalism with an embedded directional field.

## 1 INTRODUCTION

One of the most important distinctions between convolutional neural networks (CNNs) and graph neural networks (GNNs) is that CNNs allow for any convolutional kernel, while most GNN methods are limited to symmetric kernels (also called isotropic kernels in the literature) (Kipf & Welling, 2016; Xu et al., 2018a; Gilmer et al., 2017). There are some implementation of asymmetric kernels using gated mechanisms (Bresson & Laurent, 2017; Veličković et al., 2017), motif attention (Peng et al., 2019), edge features (Gilmer et al., 2017) or by using the 3D structure of molecules for message passing (Klicpera et al., 2019).

However, to the best of our knowledge, there are currently no methods that allow asymmetric graph kernels that are dependent on the full graph structure or directional flows. They either depend on local structures or local features. This is in opposition to images which exhibit canonical directions: the horizontal and vertical axes. The absence of an analogous concept in graphs makes it difficult to define directional message passing and to produce an analogue of the directional frequency filters (or Gabor filters) widely present in image processing (Olah et al., 2020).

We propose a novel idea for GNNs: use vector fields in the graph to define directions for the propagation of information, with an overview of the paper presented in 1. Hence, the aggregation or message passing will be projected onto these directions so that the contribution of each neighbouring node $n_v$ will be weighted by its alignment with the vector fields at the receiving node $n_u$. This enables our method to propagate information via directional derivatives or smoothing of the features.

We also explore using the gradients of the low-frequency eigenvectors of the Laplacian of the graph $\phi_k$, since they exhibit interesting properties (Bronstein et al., 2017; Chung et al., 1997). In particular, they can be used to define optimal partitions of the nodes in a graph, to give a natural ordering (Levy, 2006), and to find the dominant directions of the graph diffusion process (Chung & Yau, 2000). Further, we show that they generalize the horizontal and vertical directional flows in a grid (see

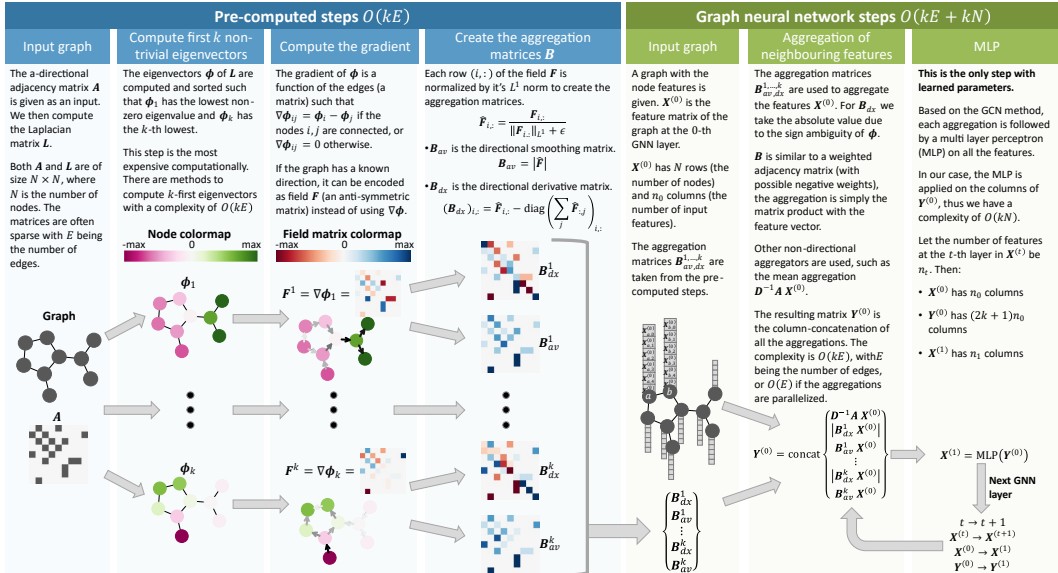

Figure 1: Overview of the steps required to aggregate messages in the direction of the eigenvectors.

figure 2), allowing them to guide the aggregation and mimic the asymmetric and directional kernels present in computer vision. In fact, we demonstrate mathematically that our work generalizes CNNs by reproducing all convolutional kernels of radius $R$ in an $n$-dimensional grid, while also bringing the powerful data augmentation capabilities of reflection, rotation or distortion of the directions.

We further show that our directional graph network (DGN) model theoretically and empirically allows for efficient message passing across distant communities, which reduces the well-known problem of over-smoothing, and aligns well with the need of independent aggregation rules (Corso et al., 2020). Alternative methods reduce the impact of over-smoothing by using skip connections (Luan et al., 2019), global pooling (Alon & Yahav, 2020), or randomly dropping edges during training time (Rong et al., 2020), but without solving the underlying problem. In fact, we also prove that DGN is more discriminative than standard GNNs in regards to the Weisfeiler-Lehman 1-WL test, showing that the reduction of over-smoothing is accompanied by an increase of expressiveness.

Our method distinguishes itself from other spectral GNNs since the literature usually uses the low frequencies to estimate local Fourier transforms in the graph (Levie et al., 2018; Xu et al., 2019). Instead, we do not try to approximate the Fourier transform, but only to define a directional flow at each node and guide the aggregation.

## 2 THEORETICAL DEVELOPMENT

### 2.1 INTUITIVE OVERVIEW

One of the biggest limitations of current GNN methods compared to CNNs is the inability to do message passing in a specific direction such as the horizontal one in a grid graph. In fact, it is difficult to define directions or coordinates based solely on the shape of the graph.

The lack of directions strongly limits the discriminative abilities of GNNs to understand local structures and simple feature transformations. Most GNNs are invariant to the permutation of the neighbours' features, so the nodes' received signal is not influenced by swapping the features of 2 neighbours. Therefore, several layers in a deep network will be employed to understand these simple changes instead of being used for higher level features, thus over-squashing the message sent between 2 distant nodes (Alon & Yahav, 2020).

In this work, one of the main contributions is the realisation that low-frequency eigenvectors of the Laplacian can overcome this limitation by providing a variety of intuitive directional flows. As a first example, taking a grid-shaped graph of size $N \times M$ with $\frac{N}{2} < M < N$, we find that the eigenvector

associated to the smallest non-zero eigenvalue increases in the direction of the width $N$ and the second one increases in the direction of the height $M$. This property generalizes to n-dimensional grids and motivated the use of gradients of eigenvectors as preferred directions for general graphs.

We validated this intuition by looking at the flow of the gradient of the eigenvectors for a variety of graphs, as shown in figure 2. For example, in the Minnesota map, the first 3 non-constant eigenvectors produce logical directions, namely South/North, suburb/city, and West/East.

Another important contribution also noted in figure 2 is the ability to define any kind of direction based on prior knowledge of the problem. Hence, instead of relying on eigenvectors to find directions in a map, we can simply use the cardinal directions or the rush-hour traffic flow.

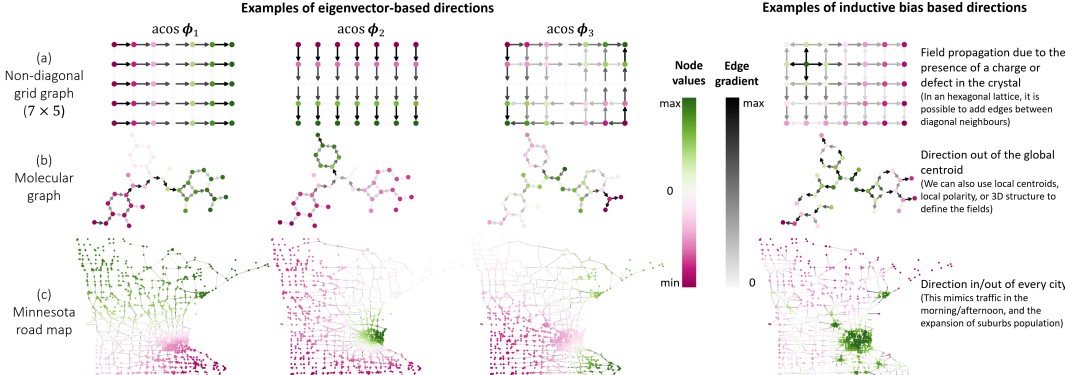

Figure 2: Possible directional flows in different types of graphs. The node coloring is a potential map and the edges represent the gradient of the potential with the arrows in the direction of the flow. The first 3 columns present the arcosine of the normalized eigenvectors (acos $\hat{\phi}$) as node coloring, and their gradients represented as edge intensity. The last column presents examples of inductive bias introduced in the choice of direction. (a) The eigenvectors 1 and 2 are the horizontal and vertical flows of the grid. (b) The eigenvectors 1 and 2 are the flow in the longest and second-longest directions. (c) The eigenvectors 1, 2 and 3 flow respectively in the South-North, suburbs to the city center and West-East directions. We ignore $\phi_0$ since it is constant and has no direction.

## 2.2 VECTOR FIELDS IN A GRAPH

Based on a recent review from Bronstein et al. (2017), this section presents the ideas of differential geometry applied to graphs, with the goal of finding proper definitions of scalar products, gradients and directional derivatives.

Let $G = (V, E)$ be a graph with $V$ the set of vertices and $E \subset V \times V$ the set of edges. The graph is undirected meaning that $(i, j) \in E$ iff $(j, i) \in E$. Define the vector spaces $L^2(V)$ and $L^2(E)$ as the set of maps $V \to \mathbb{R}$ and $E \to \mathbb{R}$ with $\boldsymbol{x}, \boldsymbol{y} \in L^2(V)$ and $\boldsymbol{F}, \boldsymbol{H} \in L^2(E)$ and scalar products

$$\langle \boldsymbol{x}, \boldsymbol{y} \rangle_{L^2(V)} := \sum_{i \in V} \boldsymbol{x}_i \boldsymbol{y}_i \quad , \qquad \langle \boldsymbol{F}, \boldsymbol{H} \rangle_{L^2(E)} := \sum_{(i,j) \in E} \boldsymbol{F}_{(i,j)} \boldsymbol{H}_{(i,j)} \tag{1}$$

Think of $E$ as the "tangent space" to $V$ and of $L^2(E)$ as the set of "vector fields" on the space $V$ with each row $\boldsymbol{F}_{i,:}$ representing a vector at the $i$-th node. Define the pointwise scalar product as the map $L^2(E) \times L^2(E) \to L^2(V)$ taking 2 vector fields and returning their inner product at each point of $V$, at the node $i$ is defined by equation 2.

$$\langle \boldsymbol{F}, \boldsymbol{H} \rangle_i := \sum_{j:(i,j) \in E} \boldsymbol{F}_{i,j} \boldsymbol{H}_{i,j} \tag{2}$$

In equation 3, we define the gradient $\nabla$ as a mapping $L^2(V) \to L^2(E)$ and the divergence div as a mapping $L^2(E) \to L^2(V)$, thus leading to an analogue of the directional derivative in equation 4.

$$(\nabla \boldsymbol{x})_{(i,j)} := \boldsymbol{x}(j) - \boldsymbol{x}(i) \quad , \qquad (\text{div } \boldsymbol{F})_i := \sum_{j:(i,j) \in E} \boldsymbol{F}_{(i,j)} \tag{3}$$

**Definition 1.** *The directional derivative of the function $\boldsymbol{x}$ on the graph $G$ in the direction of the vector field $\hat{\boldsymbol{F}}$ where each vector is of unit-norm is*

$$D_{\hat{\boldsymbol{F}}}\boldsymbol{x}(i) := \langle \nabla \boldsymbol{x}, \hat{\boldsymbol{F}} \rangle_i = \sum_{j:(i,j)\in E} (\boldsymbol{x}(j) - \boldsymbol{x}(i))\hat{\boldsymbol{F}}_{i,j} \tag{4}$$

$|\boldsymbol{F}|$ will denote the absolute value of $\boldsymbol{F}$ and $||\boldsymbol{F}_{i,:}||_{L^p}$ the $L^p$-norm of the $i$-th row of $\boldsymbol{F}$. We also define the forward/backward directions as the positive/negative parts of the field $\boldsymbol{F}^{\pm}$.

## 2.3 DIRECTIONAL SMOOTHING AND DERIVATIVES

Next, we show how the vector field $\boldsymbol{F}$ is used to *guide* the graph aggregation by projecting the incoming messages. Specifically, we define the weighted aggregation matrices $\boldsymbol{B}_{av}$ and $\boldsymbol{B}_{dx}$ that allow to compute the directional smoothing and directional derivative of the node features.

**The directional average matrix $\boldsymbol{B}_{av}$** is the weighted aggregation matrix such that all weights are positives and all rows have an $L^1$-norm equal to 1, as shown in equation 5 and theorem 2.1, with a proof in the appendix C.1.

$$\boldsymbol{B}_{av}(\boldsymbol{F})_{i,:} = \frac{|\boldsymbol{F}_{i,:}|}{||\boldsymbol{F}_{i,:}||_{L^1} + \epsilon} \tag{5}$$

The variable $\epsilon$ is an arbitrarily small positive number used to avoid floating-point errors. The $L^1$-norm denominator is a local row-wise normalization. The aggregator works by assigning a large weight to the elements in the forward or backward direction of the field, while assigning a small weight to the other elements, with a total weight of 1.

**Theorem 2.1** (Directional smoothing). *The operation $\boldsymbol{y} = \boldsymbol{B}_{av}\boldsymbol{x}$ is the directional average of $\boldsymbol{x}$, in the sense that $\boldsymbol{y}_u$ is the mean of $\boldsymbol{x}_v$, weighted by the direction and amplitude of $\boldsymbol{F}$.*

**The directional derivative matrix $\boldsymbol{B}_{dx}$** is defined in (6) and theorem 2.2, with the proof in appendix C.2. Again, the denominator is a local row-wise normalization but can be replaced by a global normalization. $\text{diag}(\boldsymbol{a})$ is a square, diagonal matrix with diagonal entries given by $\boldsymbol{a}$. The aggregator works by subtracting the projected forward message by the backward message (similar to a center derivative), with an additional diagonal term to balance both directions.

$$\boldsymbol{B}_{dx}(\boldsymbol{F})_{i,:} = \hat{\boldsymbol{F}}_{i,:} - \text{diag}\Big(\sum_j \hat{\boldsymbol{F}}_{:,j}\Big)_{i,:}, \qquad \hat{\boldsymbol{F}}_{i,:} = \left(\frac{\boldsymbol{F}_{i,:}}{||\boldsymbol{F}_{i,:}||_{L^1} + \epsilon}\right) \tag{6}$$

**Theorem 2.2** (Directional derivative). *Suppose $\hat{\boldsymbol{F}}$ have rows of unit $L^1$ norm. The operation $\boldsymbol{y} = \boldsymbol{B}_{dx}(\hat{\boldsymbol{F}})\boldsymbol{x}$ is the centered directional derivative of $\boldsymbol{x}$ in the direction of $\boldsymbol{F}$, in the sense of equation 4, i.e.*

$$\boldsymbol{y} = D_{\hat{\boldsymbol{F}}}\boldsymbol{x} = \Big(\hat{\boldsymbol{F}} - \text{diag}\Big(\sum_j \hat{\boldsymbol{F}}_{:,j}\Big)\Big)\boldsymbol{x}$$

These aggregators are directional, interpretable and complementary, making them ideal choices for GNNs. We discuss the choice of aggregators in more details in appendix A, while also providing alternative aggregation matrices such as the center-balanced smoothing, the forward-copy, the phantom zero-padding, and the hardening of the aggregators using softmax/argmax on the field. We further provide a visual interpretation of the $\boldsymbol{B}_{av}$ and $\boldsymbol{B}_{dx}$ aggregators in figure 3. Interestingly, we also note in appendix A.1 that $\boldsymbol{B}_{av}$ and $\boldsymbol{B}_{dx}$ yield respectively the mean and Laplacian aggregations when $\boldsymbol{F}$ is a vector field such that all entries are constant $\boldsymbol{F}_{ij} = \pm C$.

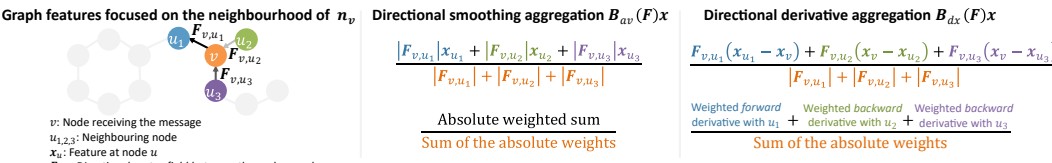

Figure 3: Illustration of how the directional aggregation works at a node $n_v$, with the arrows representing the direction and intensity of the field $\boldsymbol{F}$.

## 2.4 GRADIENT OF THE EIGENVECTORS AS INTERPRETABLE VECTOR FIELDS

In this section we give theoretical support for the choice of gradients of the eigenfunctions of the Laplacian as sensible vectors along which to do directional message passing since they are interpretable and allow to reduce the over-smoothing.

As usual the combinatorial, degree-normalized and symmetric normalized Laplacian are defined as

$$L = D - A \quad , \quad L_{\text{norm}} = D^{-1}L \quad , \quad L_{\text{sym}} = D^{-\frac{1}{2}}LD^{-\frac{1}{2}} \tag{7}$$

The problems of *over-smoothing* and *over-squashing* are critical issues in GNNs (Alon & Yahav, 2020; Hamilton, 2020). In most GNN models, node representations become over-smoothed after several rounds of message passing (i.e., convolutions), as the representations tend to reach a mean-field equilibrium equivalent to the stationary distribution of a random walk (Hamilton, 2020). Over-smoothing is also related to the problem of over-squashing, which reflects the inability for GNNs to propagate informative signals between distant nodes (Alon & Yahav, 2020) and is a major bottleneck to training deep GNN models (Xu et al., 2019). Both problems are related to the fact that the influence of one node's input on the final representation of another node in a GNN is determined by the likelihood of the two nodes co-occurring on a truncated random walk (Xu et al., 2018b).

We show in theorem 2.3 (proved in appendix C.3) that by passing information in the direction of $\phi_1$, the eigenvector associated to the lowest non-trivial frequency of $L_{\text{norm}}$, DGNs can efficiently share information between the farthest nodes of the graph, when using the *K-walk distance* to measure the difficulty of passing information. Thus, DGNs provide a natural way to address both the over-smoothing and over-squashing problems: they can efficiently propagate messages between distant nodes and in a direction that counteracts over-smoothing.

**Definition 2** (K-walk distance). *The K-walk distance $d_K(v_i, v_j)$ on a graph is the average number of times $v_i$ is hit in a K step random walk starting from $v_j$.*

**Theorem 2.3** (K-Gradient of the low-frequency eigenvectors). *Let $\lambda_i$ and $\phi_i$ be the eigenvalues and eigenvectors of the normalized Laplacian of a connected graph $L_{norm}$ and let $a, b = \arg\max_{1 \le i,j \le n}\{d_K(v_i, v_j)\}$ be the nodes that have highest K-walk distance. Let $m = \arg\min_{1 \le i \le n}(\phi_1)_i$ and $M = \arg\max_{1 \le i \le n}(\phi_1)_i$, then $d_K(v_m, v_M) - d_K(v_a, v_b)$ has order $O(1 - \lambda_2)$.*

As another point of view into the problem of oversmoothing, consider the *hitting time $Q(x, y)$* defined as the expected number of steps in a random walk starting from node $x$ ending in node $y$ with the probability transition $P(x, y) = \frac{1}{d_x}$. In appendix C.4 we give an informal argument supporting the following conjecture.

**Definition 3** (Gradient step). *Suppose the two neighboring nodes $x$ and $z$ are such that $\phi(z) - \phi(x)$ is maximal among the neighbors of $x$, then we will say $z$ is obtained from $x$ by taking a step in the direction of the gradient $\nabla\phi$.*

**Conjecture 2.4** (Gradient steps reduce expected hitting time). *Suppose that $x, y$ are uniformly distributed random nodes such that $\phi_i(x) < \phi_i(y)$. Let $z$ be the node obtained from $x$ by taking one step in the direction of $\nabla\phi_i$, then the expected hitting time is decreased proportionally to $\lambda_i^{-1}$ and*

$$\mathbb{E}_{x,y}[Q(z, y)] \le \mathbb{E}_{x,y}[Q(x, y)]$$

The next two corollaries follow from theorem 2.3 (and also conjecture 2.4 if it is true).

**Corollary 2.5** (Reduces over-squashing). *Following the direction of $\nabla\phi_1$ is an efficient way of passing information between the farthest nodes of the graph (in terms of the K-walk distance).*

**Corollary 2.6** (Reduces over-smoothing). *Following the direction of $\nabla\phi_1$ allows the influence distribution between node representations to be decorrelated from random-walk hitting times (assuming the definition of influence introduced in Xu et al. (2018b)).*

Our method also aligns perfectly with a recent proof that multiple independent aggregators are needed to distinguish neighbourhoods of nodes with continuous features (Corso et al., 2020).

When using eigenvectors of the Laplacian $\phi_i$ to define directions in a graph, we need to keep in mind that there is never a single eigenvector associated to an eigenvalue, but a whole eigenspace.

For instance, a pair of eigenvalues can have a multiplicity of 2 meaning that they can be generated by different pairs of orthogonal eigenvectors. For an eigenvalue of multiplicity 1, there are always two unit norm eigenvectors of opposite sign, which poses a problem during the directional aggregation. We can make a choice of sign and later take the absolute value (i.e. $\boldsymbol{B}_{av}$ in equation 5). An alternative is to take a sample of orthonormal basis of the eigenspace and use each choice to augment the training (see section 2.8). Although multiplicities higher than one do happen for low-frequencies (square grids have a multiplicity 2 for $\lambda_1$) this is not common in "real-world graphs"; we found no $\lambda_1$ multiplicity greater than 1 on the ZINC and PATTERN datasets (see appendix B.4). Further, although all $\phi$ are orthogonal, their gradients, used to define directions, are not always *locally* orthogonal (e.g. there are many horizontal flows in the grid). This last concern is left to be addressed in future work.

## 2.5 GENERALIZATION OF THE CONVOLUTION ON A GRID

In this section we show that our method generalizes CNNs by allowing to define any radius-$R$ convolutional kernels in grid-shaped graphs. The radius-$R$ kernel at node $u$ is a convolutional kernel that takes the weighted sum of all nodes $v$ at a distance $d(u, v) \leq R$.

Consider the lattice graph $\Gamma$ of size $N_1 \times N_2 \times ... \times N_n$ where each vertices are connected to their direct non-diagonal neighbour. We know from Lemma C.1 that, for each dimension, there is an eigenvector that is only a function of this specific dimension. For example, the lowest frequency eigenvector $\phi_1$ always flows in the direction of the longest length. Hence, the Laplacian eigenvectors of the grid can play a role analogous to the axes in Euclidean space, as shown in figure 2.

With this knowledge, we show in theorem 2.7 (proven in C.7), that we can generalize all convolutional kernels in an n-dimensional grid. This is a strong result since it demonstrates that our DGN framework generalizes CNNs when applied on a grid, thus closing the gap between GNNs and the highly successful CNNs on image tasks.

**Theorem 2.7** (Generalization radius-$R$ convolutional kernel in a lattice). *For an $n$-dimensional lattice, any convolutional kernel of radius $R$ can be realized by a linear combination of directional aggregation matrices and their compositions.*

As an example, figure 4 shows how a linear combination of the first and $m$-th aggregators $\boldsymbol{B}(\nabla\phi_{1,m})$ realize a kernel on an $N \times M$ grid, where $m = \lceil N/M \rceil$ and $N > M$.

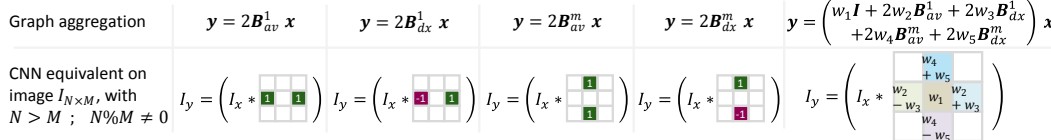

Figure 4: Realization of a radius-1 convolution using the proposed aggregators. $I_x$ is the input feature map, $*$ the convolutional operator, $I_y$ the convolution result, and $\boldsymbol{B}^i = \boldsymbol{B}(\nabla\phi_i)$.

## 2.6 EXTENDING THE RADIUS OF THE AGGREGATION KERNEL

Having aggregation kernels for neighbours of distance 2 or 3 is important to improve the expressiveness of GNNs, their ability to understand patterns, and to reduce the number of layers required. However, the lack of directions in GNNs strongly limits the radius of the kernels since, given a graph of regular degree $d$, a mean/sum aggregation at a radius-$R$ will result in a heavy over-squashing of $d^R$ messages. Using the directional fields, we can enumerate different paths, thus assigning a different weight for different $R$-distant neighbours. This method, proposed in appendix A.7, avoids the over-squashing, but empirical results are left for future work.

## 2.7 COMPARISON WITH WEISFEILER-LEHMAN (WL) TEST

We also compare the expressiveness of the Directional Graph Networks with the classical WL graph isomorphism test which is often used to classify the expressivity of graph neural networks (Xu et al., 2018a). In theorem 2.8 (proven in appendix C.8) we prove that DGNs are capable of distinguishing pairs of graphs that the 1-WL test (and so ordinary GNNs) cannot differentiate.

**Theorem 2.8** (Comparison with 1-WL test). *DGNs using the mean aggregator, any directional aggregator of the first eigenvector and injective degree-scalers are strictly more powerful than the 1-WL test.*

## 2.8 DATA AUGMENTATION

Another important result is that the directions in the graph allow to replicate some of the most common data augmentation techniques used in computer vision, namely reflection, rotation and distortion. The main difference is that, instead of modifying the image (such as a $5°$ rotation), the proposed transformation is applied on the vector field defining the aggregation kernel (thus rotating the kernel by $-5°$ without changing the image). This offers the advantage of avoiding to pre-process the data since the augmentation is done directly on the kernel at each iteration of the training.

The simplest augmentation is the vector field flipping, which is done changing the sign of the field $\boldsymbol{F}$, as stated in definition 4. This changes the sign of $\boldsymbol{B}_{dx}$, but leaves $\boldsymbol{B}_{av}$ unchanged.

**Definition 4** (Reflection of the vector field). *For a vector field $\boldsymbol{F}$, the reflected field is $-\boldsymbol{F}$.*

Let $\boldsymbol{F}_1, \boldsymbol{F}_2$ be vector fields in a graph, with $\hat{\boldsymbol{F}}_1$ and $\hat{\boldsymbol{F}}_2$ being the field normalized such that each row has a unitary $L^2$-norm. Define the angle vector $\boldsymbol{\alpha}$ by $\langle(\hat{\boldsymbol{F}}_1)_{i,:}, (\hat{\boldsymbol{F}}_2)_{i,:}\rangle = \cos(\boldsymbol{\alpha}_i)$. The vector field $\hat{\boldsymbol{F}}_2^{\perp}$ is the normalized component of $\hat{\boldsymbol{F}}_2$ perpendicular to $\hat{\boldsymbol{F}}_1$. The equation below defines $\hat{\boldsymbol{F}}_2^{\perp}$. The next equation defines the angle

$$(\hat{\boldsymbol{F}}_2^{\perp})_{i,:} = \frac{(\hat{\boldsymbol{F}}_2 - \langle\hat{\boldsymbol{F}}_1, \hat{\boldsymbol{F}}_2\rangle\hat{\boldsymbol{F}}_1)_{i,:}}{||(\hat{\boldsymbol{F}}_2 - \langle\hat{\boldsymbol{F}}_1, \hat{\boldsymbol{F}}_2\rangle\hat{\boldsymbol{F}}_1)_{i,:}||}$$

Notice that we then have the decomposition $(\hat{\boldsymbol{F}}_2)_{i,:} = \cos(\boldsymbol{\alpha}_i)(\hat{\boldsymbol{F}}_1)_{i,:} + \sin(\boldsymbol{\alpha}_i)(\hat{\boldsymbol{F}}_2^{\perp})_{i,:}$.

**Definition 5** (Rotation of the vector fields). *For $\hat{\boldsymbol{F}}_1$ and $\hat{\boldsymbol{F}}_2$ non-colinear vector fields with each vector of unitary length, their rotation by the angle $\theta$ in the plane formed by $\{\hat{\boldsymbol{F}}_1, \hat{\boldsymbol{F}}_2\}$ is*

$$\hat{\boldsymbol{F}}_1^{\theta} = \hat{\boldsymbol{F}}_1\mathrm{diag}(\cos\theta) + \hat{\boldsymbol{F}}_2^{\perp}\mathrm{diag}(\sin\theta) \quad , \quad \hat{\boldsymbol{F}}_2^{\theta} = \hat{\boldsymbol{F}}_1\mathrm{diag}(\cos(\theta+\boldsymbol{\alpha})) + \hat{\boldsymbol{F}}_2^{\perp}\mathrm{diag}(\sin(\theta+\boldsymbol{\alpha})) \quad (8)$$

Finally, the following augmentation has a similar effect to a wave distortion applied on images.

**Definition 6** (Random distortion of the vector field). *For vector field $\boldsymbol{F}$ and anti-symmetric random noise matrix $\boldsymbol{R}$, its randomly distorted field is $\boldsymbol{F}' = \boldsymbol{F} + \boldsymbol{R} \circ \boldsymbol{A}$.*

## 3 IMPLEMENTATION

We implemented the models using the DGL and PyTorch libraries and we provide the code at the address https://anonymous.4open.science/r/a752e2b1-22e3-40ce-851c-a564073e1fca/. We test our method on standard benchmarks from Dwivedi et al. (2020) and Hu et al. (2020), namely ZINC, CIFAR10, PATTERN and MolHIV with more details on the datasets and how we enforce a fair comparison in appendix B.1.

For the empirical experiments we inserted our proposed aggregation method in two different type of message passing architecture used in the literature: a *simple* one similar to the one present in GCN (equation 9a) (Kipf & Welling, 2016) and a more *complex* and general one typical of MPNN (9b) (Gilmer et al., 2017) with or without edge features $e_{ji}$. Hence, the time complexity $O(Em)$ is identical to the PNA (Corso et al., 2020), where $E$ is the number of edges and $m$ the number of aggregators, with an additional $O(Ek)$ to pre-compute the $k$-first eigenvectors, as explained in the appendix B.2.

$$X_i^{(t+1)} = U\left(\bigoplus_{(j,i)\in E} X_j^{(t)}\right) \quad (9a) \qquad X_i^{(t+1)} = U\left(X_i^{(t)}, \bigoplus_{(j,i)\in E} M\left(X_i^{(t)}, X_j^{(t)}, \underbrace{e_{ji}}_{\text{optional}}\right)\right) \quad (9b)$$

where $\bigoplus$ is an operator which concatenates the results of multiple aggregators, $X$ is the node features, $M$ is a linear transformation and $U$ a multiple layer perceptron.

We tested the directional aggregators across the datasets using the gradient of the first $k$ eigenvectors $\nabla\phi_{1,...,k}$ as the underlying vector fields. Here, $k$ is a hyperparameter, usually 1 or 2, but could be bigger for high-dimensional graphs. To deal with the arbitrary sign of the eigenvectors, we take the absolute value of the result of equation 6, making it invariant to a reflection of the field. In case of a disconnected graph, $\phi_i$ is the $i$-th eigenvector of each connected component. Despite the numerous aggregators proposed in appendix A, only $\boldsymbol{B}_{dx}$ and $\boldsymbol{B}_{av}$ are tested empirically.

## 4 RESULTS AND DISCUSSION

**Directional aggregation**    Using the benchmarks introduced in section 3, we present in figure 5 a fair comparison of various aggregation strategies using the same parameter budget and hyperparameters. We see a consistent boost in the performance for *simple*, *complex* and *complex with edges* models using directional aggregators compared to the *mean-aggregator* baseline.

| Aggregators | ZINC | | | PATTERN | | CIFAR10 | | MolHIV | |
| --- | --- | --- | --- | --- | --- | --- | --- | --- | --- |
| | Simple | Complex | Complex-E | Simple | Complex | Simple | Complex | Simple | |
| | MAE | MAE | MAE | % acc | % acc | % acc | % acc | % ROC-AUC | Best |
| mean | 0.316 | 0.353 | 0.262 | 80.77 | 83.34 | 55.9 | 62.8 | 75.1 | |
| mean pos$_1$ | 0.349 | 0.332 | 0.297 | 80.76 | 83.74 | | | 75.8 | |
| mean pos$_1$ pos$_2$ | 0.344 | 0.330 | 0.284 | 84.51 | 81.25 | | | 76.1 | |
| mean dx$_1$ | 0.296 | **0.233** | **0.191** | 84.22 | 83.44 | | | 78.0 | |
| mean dx$_1$ dx$_2$ | 0.337 | 0.271 | 0.205 | 81.61 | 86.62 | 52.9 | **69.8** | 76.5 | |
| mean av$_1$ | 0.317 | 0.332 | 0.276 | 84.54 | 83.21 | | | 78.4 | Worst |
| mean av$_1$ av$_2$ | 0.367 | 0.332 | 0.260 | 85.12 | 85.38 | **60.6** | 65.1 | 77.1 | |
| mean dx$_1$ av$_1$ | **0.290** | 0.245 | 0.192 | **85.17** | **86.68** | | | **79.0** | |

Figure 5: Test set results using a parameter budget of $\sim 100k$, with the same hyperparameters as Corso et al. (2020). The low-frequency Laplacian eigenvectors are used to define the directions, except for CIFAR10 that uses the coordinates of the image. For brevity, we denote $dx_\mathrm{i}$ and $av_\mathrm{i}$ as the directional derivative $\boldsymbol{B}_{dx}^i$ and smoothing $\boldsymbol{B}_{av}^i$ aggregators of the $i$-th direction. We also denote $pos_\mathrm{i}$ as the $i$-th eigenvector used as positional encoding for the mean aggregator.

In particular, we see a significant improvement in ZINC and MolHIV using the directional aggregators. We believe this is due to the capacity to move efficiently messages across opposite parts of the molecule and to better understand the role of atom pairs. Further, the thesis that DGNs can bridge the gap between CNNs and GNNs is supported by the clear improvements on CIFAR10 over the baselines. This contrasts with the positional encoding which showed no clear improvement.

With our theoretical analysis in mind, we expected to perform well on PATTERN since the flow of the first eigenvectors are meaningful directions in a stochastic block model and passing messages using those directions allows the network to efficiently detect the two communities. The results match our expectations, outperforming all the previous models.

**Comparison to the literature**    In order to compare our model with the literature, we fine-tuned it on the various datasets and we report its performance in figure 6. We observe that DGN provides significant improvement across all benchmarks, highlighting the importance of anisotropic kernels. In the work by Dwivedi et al. (2020), they proposed the use of positional encoding of the eigenvectors in node features, but these bring significant improvement when many eigenvectors and high network depths are used. Our results outperform them with fewer parameters, less depth, and only 1-2 eigenvectors, further motivating their use as directional flows instead of positional encoding.

**Data augmentation**    To evaluate the effectiveness of the proposed augmentation, we trained the models on a reduced version of the CIFAR10 dataset. The results in figure 7 show clearly a higher expressive power of the $dx$ aggregator, enabling it to fit well the training data. For a small dataset, this comes at the cost of overfitting and a reduced test-set performance, but we observe that randomly rotating or distorting the kernels counteracts the overfitting and improves the generalization.

As expected, the performance decreases when the rotation or distortion is too high since the augmented graph changes too much. In computer vision images similar to CIFAR10 are usually rotated by less than $30°$ (Shorten & Khoshgoftaar; O'Gara & McGuinness, 2019). Further, due to the constant number of parameters across models, less parameters are attributed to the mean aggregation in

| Model | ZINC | | PATTERN | CIFAR10 | | MolHIV |
| --- | --- | --- | --- | --- | --- | --- |
| | No edge features | Edge features | No edge features | No edge features | Edge features | No edge features |
| | MAE | MAE | % acc | % acc | % acc | % ROC-AUC |
| GCN | $0.469_{\pm0.002}$ | | $65.880_{\pm0.074}$ | $54.46_{\pm0.10}$ | | $76.06_{\pm0.97}$ * |
| GIN | $0.408_{\pm0.008}$ | | $85.590_{\pm0.011}$ | $53.28_{\pm3.70}$ | | $75.58_{\pm1.40}$ * |
| GraphSage | $0.410_{\pm0.005}$ | | $50.516_{\pm0.001}$ | $66.08_{\pm0.24}$ | | |
| GAT | $0.463_{\pm0.002}$ | | $75.824_{\pm1.823}$ | $65.48_{\pm0.33}$ | | |
| MoNet | $0.407_{\pm0.007}$ | | $85.482_{\pm0.037}$ | $53.42_{\pm0.43}$ | | |
| GatedGCN | $0.422_{\pm0.006}$ | $0.363_{\pm0.009}$ | $84.480_{\pm0.122}$ | $69.19_{\pm0.28}$ | $69.37_{\pm0.48}$ | |
| PNA | $0.320_{\pm0.032}$ | $0.188_{\pm0.004}$ | $86.567_{\pm0.075}$ | $70.46_{\pm0.44}$ | $70.47_{\pm0.72}$ | $79.05_{\pm1.32}$ * |
| DGN | $0.219_{\pm0.010}$ | $0.168_{\pm0.003}$ | $86.680_{\pm0.034}$ | $72.70_{\pm0.54}$ | $72.84_{\pm0.42}$ | $79.70_{\pm0.97}$ |

Figure 6: Fine-tuned results of the DGN model against models from Dwivedi et al. (2020) and Hu et al. (2020): GCN (Kipf & Welling, 2016), GraphSage (Hamilton et al., 2017), GIN (Xu et al., 2018a), GAT (Veličković et al., 2017), MoNet (Monti et al., 2017), GatedGCN (Bresson & Laurent, 2017) and PNA (Corso et al., 2020). All the models use $\sim 100k$ parameters, except those with * who use $300k$ to $1.9M$. In ZINC the DGN aggregators are {*mean, $dx_1$, max, min*}, in PATTERN {*mean, $dx_1$, $av_1$*}, in CIFAR10 {*mean, $dx_1$, $dx_2$, max*}, in MolHIV {*mean, $dx_1$, $av_1$, max, min*}.

the directional models, thus it cannot fit well the data when the rotation/distortion is too strong since the directions are less informative. We expect large models to perform better at high angles.

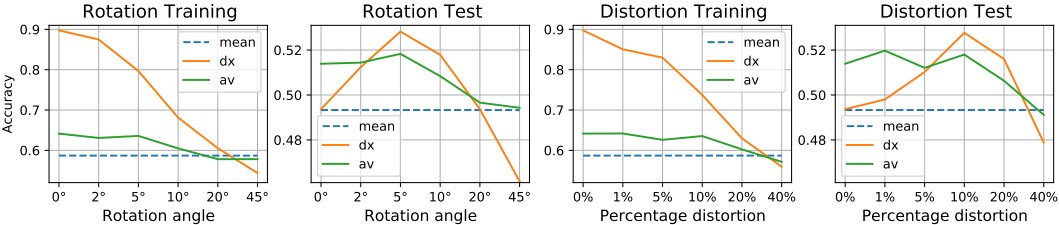

Figure 7: Accuracy of the various models using data augmentation with a *complex* architecture of $\sim 100k$ parameters and trained on 10% of the CIFAR10 training set (4.5k images). An angle of $x$ corresponds to a rotation of the kernel by a random angle sampled uniformly in $(-x^\circ, x^\circ)$ using definition 5 with $F_{1,2}$ being the gradient of the horizontal/vertical coordinates. A noise of $100x\%$ corresponds to a distortion of each eigenvector with a random noise uniformly sampled in $(-x \cdot m, x \cdot m)$ where $m$ is the average absolute value of the eigenvector's components. The *mean* baseline model is not affected by the augmentation since it does not use the underlining vector field.

## 5 CONCLUSION

The proposed DGN method allows to solve many problems of GNNs, including the lack of anisotropy, the low expressiveness, the over-smoothing and over-squashing. For the first time in graph networks, we generalize the directional properties of CNNs and their data augmentation capabilities. Based on an intuitive idea and backed by a set of strong theoretical and empirical results, we believe this work will give rise to a new family of directional GNNs. Future work can focus on the implementation of radius-$R$ kernels and improving the choice of multiple orthogonal directions.

**Broader Impact** This work will extend the usability of graph networks to all problems with physically defined directions, thus making GNN a new laboratory for physics, material science and biology. In fact, the anisotropy present in a wide variety of systems could be expressed as vector fields (spinor, tensor) compatible with the DGN framework, without the need of eigenvectors. One example is magnetic anisotropicity in metals, alloys and also in molecules such as benzene ring, alkene, carbonyl, alkyne that are easier or harder to magnetise depending on the directions or which way the object is rotated. Other examples are the response of materials to high electromagnetic fields (e.g. to study material responses at terahertz frequency); all kind of field propagation in crystals lattices (vibrations, heat, shear and frictional force, young modulus, light refraction, birefringence); multi-body or liquid motion; traffic modelling; and design of novel materials and constrained structures. This also enables GNNs to be used for virtual prototyping systems since the added directional constraints could improve the analysis of a product's functionality, manufacturing and behavior.

AUTHOR CONTRIBUTIONS

Anonymous

ACKNOWLEDGMENTS

Anonymous

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

## A    APPENDIX - CHOICES OF DIRECTIONAL AGGREGATORS

This appendix helps understand the choice of $\boldsymbol{B}_{av}$ and $\boldsymbol{B}_{dx}$ in section 2.3 and presents different directional aggregators that can be used as an alternative to the ones proposed.

A simple alternative to the directional smoothing and directional derivative operator is to simply take the *forward/backward* values according to the underlying positive/negative parts of the field $\boldsymbol{F}$, since it can effectively replicate them. However, there are many advantage of using $\boldsymbol{B}_{av,dx}$. First, one can decide to use either of them and still have an interpretable aggregation with half the parameters. Then, we also notice that $\boldsymbol{B}_{av,dx}$ regularize the parameter by forcing the network to take both forward and backward neighbours into account at each time, and avoids one of the neighbours becoming too important. Lastly, they are robust to a change of sign of the eigenvectors since $\boldsymbol{B}_{av}$ is sign invariant and $\boldsymbol{B}_{dx}$ will only change the sign of the results, which is not the case for *forward/backward* aggregations.

### A.1    RETRIEVING THE MEAN AND LAPLACIAN AGGREGATIONS

It is interesting to note that we can recover simple aggregators from the aggregation matrices $\boldsymbol{B}_{av}(\boldsymbol{F})$ and $\boldsymbol{B}_{dx}(\boldsymbol{F})$. Let $\boldsymbol{F}$ be a vector field such that all edges are equally weighted $\boldsymbol{F}_{ij} = \pm C$ for all edges $(i, j)$. Then, the aggregator $\boldsymbol{B}_{av}$ is equivalent to a mean aggregation:

$$\boldsymbol{B}_{av}(\boldsymbol{F})\boldsymbol{x} = \boldsymbol{D}^{-1}\boldsymbol{A}\boldsymbol{x}$$

Under the condition $F_{ij} = C$, the differential aggregator is equivalent to a Laplacian operator $\boldsymbol{L}$ normalized using the degree $\boldsymbol{D}$

$$\boldsymbol{B}_{dx}(C\boldsymbol{A})\boldsymbol{x} = \boldsymbol{D}^{-1}(\boldsymbol{A} - \boldsymbol{D})\boldsymbol{x} = -\boldsymbol{D}^{-1}\boldsymbol{L}\boldsymbol{x}$$

### A.2    GLOBAL FIELD NORMALIZATION

The proposed aggregators are defined with a row-wise normalized field

$$\hat{\boldsymbol{F}_{i,:}} = \frac{\boldsymbol{F}_{i,:}}{||\boldsymbol{F}_{i,:}||_{L^P}}$$

meaning that all the vectors are of unit-norm and the aggregation/message passing is done only according to the direction of the vectors, not their amplitude. However, it is also possible to do a global normalization of the field $\boldsymbol{F}$ by taking a matrix-norm instead of a vector-norm. Doing so will modulate the aggregation by the amplitude of the field at each node. One needs to be careful since a global normalization might be very sensitive to the number of nodes in the graph.

### A.3    CENTER-BALANCED AGGREGATORS

A problem arises in the aggregators $\boldsymbol{B}_{dx}$ and $\boldsymbol{B}_{av}$ proposed in equations 5 and 6 when there is an imbalance between the positive and negative terms of $\boldsymbol{F}^{\pm}$. In that case, one of the directions overtakes the other in terms of associated weights.

An alternative is also to normalize the forward and backward directions separately, to avoid having either the backward or forward direction dominating the message.

$$\boldsymbol{B}_{av-center}(\boldsymbol{F})_{i,:} = \frac{\boldsymbol{F}_{i,:}'^{+} + \boldsymbol{F}_{i,:}'^{-}}{||\boldsymbol{F}_{i,j}'^{+} + \boldsymbol{F}_{i,j}'^{-}||_{L_1}} \quad , \qquad \boldsymbol{F}_{i,:}'^{\pm} = \frac{|\boldsymbol{F}_{i,:}^{\pm}|}{||\boldsymbol{F}_{i,:}^{\pm}||_{L^1} + \epsilon} \tag{10}$$

The same idea can be applied to the derivative aggregator equation 11 where the positive and negative parts of the field $\boldsymbol{F}^{\pm}$ are normalized separately to allow to project both the *forward* and *backward* messages into a vector field of unit-norm. $\boldsymbol{F}^{+}$ is the out-going field at each node and is used for the *forward* direction, while $\boldsymbol{F}^{-}$ is the in-going field used for the *backward* direction. By averaging the *forward* and *backward* derivatives, the proposed matrix $\boldsymbol{B}_{dx\text{-center}}$ represents the centered derivative matrix.

$$\boldsymbol{B}_{dx\text{-center}}(\boldsymbol{F})_{i,:} = \boldsymbol{F}_{i,:}' - \text{diag}\left(\sum_j \boldsymbol{F}_{:,j}'\right)_{i,:} \quad , \quad \boldsymbol{F}_{i,:}' = \frac{1}{2}\left(\underbrace{\frac{\boldsymbol{F}_{i,:}^{+}}{||\boldsymbol{F}_{i,:}^{+}||_{L^1} + \epsilon}}_{\text{forward field}} + \underbrace{\frac{\boldsymbol{F}_{i,:}^{-}}{||\boldsymbol{F}_{i,:}^{-}||_{L^1} + \epsilon}}_{\text{backward field}}\right) \tag{11}$$

## A.4 HARDENING THE AGGREGATORS

The aggregation matrices that we proposed, mainly $\boldsymbol{B}_{dx}$ and $\boldsymbol{B}_{av}$ depend on a smooth vector field $\boldsymbol{F}$. At any given node, the aggregation will take a weighted sum of the neighbours in relation to the direction of $\boldsymbol{F}$. Hence, if the field $\boldsymbol{F}_v$ at a node $v$ is *diagonal* in the sense that it gives a non-zero weight to many neighbours, then the aggregator will compute a weighted average of the neighbours.

Although there are clearly good reasons to have this weighted-average behaviour, it is not necessarily desired in every problem. For example, if we want to move a single node across the graph, this behaviour will smooth the node at every step. Instead, we propose below to soften and harden the aggregations by forcing the field into making a decision on the direction it takes.

**Soft hardening the aggregation** is possible by using a softmax with a temperature $T$ on each row to obtain the field $\boldsymbol{F}_{\text{softhard}}$.

$$(\boldsymbol{F}_{\text{softhard}})_{i,:} = \text{sign}(\boldsymbol{F}_{i,:})\text{softmax}(T|\boldsymbol{F}_{i,:}|) \tag{12}$$

**Hardening the aggregation** is possible by using an infinite temperature, which changes the softmax functions into argmax. In this specific case, the node with the highest component of the field will be copied, while all other nodes will be ignored.

$$(\boldsymbol{F}_{\text{hard}})_{i,:} = \text{sign}(\boldsymbol{F}_{i,:})\text{argmax}(|\boldsymbol{F}_{i,:}|) \tag{13}$$

An alternative to the aggregators above is to take the *softmin/argmin* of the negative part and the *softmax/argmax* of the positive part.

## A.5 FORWARD AND BACKWARD COPY

The aggregation matrices $\boldsymbol{B}_{av}$ and $\boldsymbol{B}_{dx}$ have the nice property that if the field is flipped (change of sign), the aggregation gives the same result, except for the sign of $\boldsymbol{B}_{dx}$. However, there are cases where we want to propagate information in the forward direction of the field, without smoothing it with the backward direction. In this case, we can define the strictly forward and strictly backward fields below, and use them directly with the aggregation matrices.

$$\boldsymbol{F}_{\text{forward}} = \boldsymbol{F}^{+} \quad , \qquad \boldsymbol{F}_{\text{backward}} = \boldsymbol{F}^{-} \tag{14}$$

Further, we can use the hardened fields in order to define a forward copy and backward copy, which will simply copy the node in the direction of the highest field component.

$$\boldsymbol{F}_{\text{forward copy}} = \boldsymbol{F}_{\text{hard}}^{+} \quad , \qquad \boldsymbol{F}_{\text{backward copy}} = \boldsymbol{F}_{\text{hard}}^{-} \tag{15}$$

## A.6 Phantom zero-padding

Some recent work in computer vision has shown the importance of zero-padding to improve CNNs by allowing the network to understand it's position relative to the border (Islam et al., 2020). In contrast, using boundary conditions or reflection padding makes the network completely blind to positional information. In this section, we show that we can mimic the zero-padding in the direction of the field $F$ for both aggregation matrices $B_{av}$ and $B_{dx}$.

Starting with the $B_{av}$ matrix, in the case of a missing neighbour in the forward/backward direction, the matrix will compensate by adding more weights to the other direction, due to the denominator which performs a normalization. Instead, we would need the matrix to consider both directions separately so that a missing direction would result in zero padding. Hence, we define $B_{av,0pad}$ below, where either the $F^+$ or $F^-$ will be 0 on a boundary with strictly in-going/out-going field.

$$(B_{av,0pad})_{i,:} = \frac{1}{2}\left(\frac{|F_{i,:}^+|}{||F_{i,:}^+||_{L^1} + \epsilon} + \frac{|F_{i,:}^-|}{||F_{i,:}^-||_{L^1} + \epsilon}\right) \tag{16}$$

Following the same argument, we define $B_{dx,0pad}$ below, where either the forward or backward term is ignored. The diagonal term is also removed at the boundary so that the result is a center derivative equal to the subtraction of the forward term with the 0-term on the back (or vice-versa), instead of a forward derivative.

$$B_{dx-0pad}(F)_{i,:} = \begin{cases} F_{i,:}'^+ & \text{if } \sum_j F_{i,j}'^- = 0 \\ F_{i,:}'^- & \text{if } \sum_j F_{i,j}'^+ = 0 \\ \frac{1}{2}\left(F_{i,:}'^+ + F_{i,:}'^- - \text{diag}\left(\sum_j F_{:,j}'^+ + F_{:,j}'^-\right)_{i,:}\right), & \text{otherwise} \end{cases} \tag{17}$$

$$F_{i,:}'^+ = \frac{F_{i,:}^+}{||F_{i,:}^+||_{L^1} + \epsilon} \qquad F_{i,:}'^- = \frac{F_{i,:}^-}{||F_{i,:}^-||_{L^1} + \epsilon}$$

## A.7 Extending the radius of the aggregation kernel

We aim at providing a general radius-$R$ kernel $B_R$ that assigns different weights to different subsets of nodes $n_u$ at a distance $R$ from the center node $n_v$.

First, we decompose the matrix $B(F)$ into positive and negative parts $B^\pm(F)$ representing the forward and backward steps aggregation in the field $F$.

$$B(F) = B^+(F) - B^-(F) \tag{18}$$

Thus, defining $B_{fb}^\pm(F)_{i,:} = \frac{F_{i,:}^\pm}{||F_{i,:}||_{L^p}}$, we can find different aggregation matrices by using different combinations of walks of radius $R$. First demonstrated for a grid in theorem 2.7, we generalize it in equation 19 for any graph $G$.

**Definition 7** (General radius $R$ n-directional kernel). *Let $S_n$ be the group of permutations over $n$ elements with a set of directional fields $F_i$.*

$$B_R := \underbrace{\sum_{\substack{V=\{v_1,v_2,...,v_n\}\in\mathbb{N}^n \\ ||V||_{L^1}\leq R, \quad -R\leq v_i\leq R}}}_{\substack{\text{Any choice of walk } V \text{ with at most } R \text{ steps} \\ \text{using all combinations of } v_1, v_2, ..., v_n}} \underbrace{\sum_{\sigma\in S_n} a_V}_{\substack{\text{optional} \\ \text{permutations}}} \underbrace{\prod_{j=1}^N (B_{fb}^{sgn(v_{\sigma(j)})}(F_{\sigma(j)}))^{|v_{\sigma(j)}|}}_{\text{Aggregator following the steps } V, \text{ permuted by } S_n} \tag{19}$$

In this equation, $n$ is the number of directional fields and $R$ is the desired radius. $V$ represents all the choices of walk $\{v_1, v_2, ..., v_n\}$ in the direction of the fields $\{F_1, F_2, ..., F_n\}$. For example, $V = \{3, 1, 0, -2\}$ has a radius $R = 6$, with 3 steps *forward* of $F_1$, 1 step *forward* of $F_2$, and 2 steps *backward* of $F_4$. The sign of each $B_{fb}^\pm$ is dependant to the sign of $v_{\sigma(j)}$, and the power $|v_{\sigma(j)}|$ is the

number of aggregation steps in the directional field $F_{\sigma(j)}$. The full equation is thus the combination of all possible choices of paths across the set of fields $F_i$, with all possible permutations. Note that we are restricting the sum to $v_i$ having only a possible sign; although matrices don't commute, we avoid choosing different signs since it will likely self-intersect a lower radius walk. The permutations $\sigma$ are required since, for example, the path *up* $\rightarrow$ *left* is different (in a general graph) than the path *left* $\rightarrow$ *up*.

This matrix $B_R$ has a total of $\sum_{r=0}^{R}(2n)^r = \frac{(2n)^{R+1}-1}{2n-1}$ parameters, with a high redundancy since some permutations might be very similar, e.g. for a grid graph we have that *up* $\rightarrow$ *left* is identical to *left* $\rightarrow$ *up*. Hence, we can replace the permutation $S_n$ by a reverse ordering, meaning that $\prod_{j}^{N} B_j = B_N...B_2 B_1$. Doing so does not perfectly generalize the radius-$R$ kernel for all graphs, but it generalizes it on a grid and significantly reduces the number of parameters to $\sum_{r=0}^{R} \sum_{l=1}^{min(n,r)} 2^r \binom{n}{l}\binom{r-1}{l-1}$.

# B  APPENDIX - IMPLEMENTATION DETAILS

## B.1  BENCHMARKS AND DATASETS

We use a variety of benchmarks proposed by Dwivedi et al. (2020) and Hu et al. (2020) to test the empirical performance of our proposed methods. In particular, to have a wide variety of graphs and tasks we chose:

1. ZINC, a graph regression dataset from molecular chemistry. The task is to predict a score that is a subtraction of computed properties $logP - SA$, with $logP$ being the computed octanol-water partition coefficient, and $SA$ being the synthetic accessibility score (Jin et al., 2018).

2. CIFAR10, a graph classification dataset from computer vision (Krizhevsky, 2009). The task is to classify the images into 10 different classes, with a total of 5000 training image per class and 1000 test image per class. Each image has $32 \times 32$ pixels, but the pixels have been clustered into a graph of $\sim 100$ super-pixels. Each super-pixel becomes a node in an *almost* grid-shaped graph, with 8 edges per node. The clustering uses the code from Knyazev et al. (2019), and results in a different number of super-pixels per graph.

3. PATTERN, a node classification synthetic benchmark generated with Stochastic Block Models, which are widely used to model communities in social networks. The task is to classify the nodes into 2 communities and it tests the fundamental ability of recognizing specific predetermined subgraphs.

4. MolHIV, a graph classification benchmark from molecular chemistry. The task is to predict whether a molecule inhibits HIV virus replication or not. The molecules in the training, validation and test sets are divided using a scaffold splitting procedure that splits the molecules based on their two-dimensional structural frameworks.

Our goal is to provide a fair comparison to demonstrate the capacity of our proposed aggregators. Therefore, we compare the various methods on both types of architectures using the same hyper-parameters tuned in previous works (Corso et al., 2020) for similar networks. The models vary exclusively in the aggregation method and the width of the architectures to keep a set parameter budget.

In CIFAR10 it is impossible to numerically compute a deterministic vector field with eigenvectors due to the multiplicity of $\lambda_1$ being greater than 1. This is caused by the symmetry of the square image, and is extremely rare in real-world graphs. Therefore, we used as underlying vector field the gradient of the coordinates of the image. Note that these directions are provided in the nodes' features in the dataset and available to all models, that they are co-linear to the eigenvectors of the grid as per lemma C.1, and that they mimic the inductive bias in CNNs.

## B.2  IMPLEMENTATION AND COMPUTATIONAL COMPLEXITY

Unlike several more expressive graph networks (Kondor et al., 2018; Maron et al., 2018), our method does not require a computational complexity superlinear with the size of the graph. The calculation

of the first $k$ eigenvectors during pretraining, done using Lanczos method (Lanczos, 1950) and the sparse module of Scipy, has a time complexity of $O(Ek)$ where $E$ is the number of edges. During training the complexity is equivalent to a $m$-aggregator GNN $O(Em)$ (Corso et al., 2020) for the aggregation and $O(Nm)$ for the MLP.

To all the architectures we added residual connections (He et al., 2016), batch normalization (Ioffe & Szegedy, 2015) and graph size normalization (Dwivedi et al., 2020).

For all the datasets with non-regular graphs, we combine the various aggregators with logarithmic degree-scalers as in Corso et al. (2020).

An important thing to note is that, for dynamic graphs, the eigenvectors need to be re-computed dynamically with the changing edges. Fortunately, there are random walk based algorithms that can estimate $\phi_1$ quickly, especially for small changes to the graph (Doshi & Eun, 2000). In the current empirical results, we do not work with dynamic graphs.

## B.3 RUNNING TIME

The precomputation of the first four eigenvectors for all the graphs in the datasets takes $38s$ for ZINC, $96s$ for PATTERN and $120s$ for MolHIV on CPU. Table 1 shows the average running time on GPU for all the various model from figure 5. On average, the epoch running time is 16% slower for the DGN compared to the mean aggregation, but a faster convergence for DGN means that the total training time is on average 8% faster for DGN.

Table 1: Average running time for the non-fine tuned models from figure 5. Each entry represents average time per epoch / average total training time. Each of these models has a parameter budget $\sim 100k$ and was run on a Tesla T4 (15GB GPU). The *avg increase* row is the average of the relative running time of all rows compared to the *mean* row, with a negative value meaning a faster running time.

| Aggregators | ZINC | | | PATTERN | |
|---|---|---|---|---|---|
| | Simple | Complex | Complex-E | Simple | Complex |
| mean | 3.29s/1505s | 3.58s/1584s | 3.56s/1654s | 153.1s/10154s | 117.8s/9031s |
| mean $dx_1$ | 3.86s/1122s | 3.77s/1278s | 4.22s/1371s | 144.9s/8109s | 127.2s/8417s |
| mean $dx_1$ $dx_2$ | 4.23s/1360s | 4.55s/1560s | 4.63s/1680s | 153.3s/8057s | 167.9s/9326s |
| mean $av_1$ | 3.68s/1297s | 3.84s/1398s | 3.92s/1272s | 128.0s/8680s | 88.1s/7456s |
| mean $av_1$ $av_2$ | 3.95s/1432s | 4.03s/1596s | 4.07s/1721s | 134.2s/8115s | 170.4s/11114s |
| mean $dx_1$ $av_1$ | 3.89s/1079s | 4.09s/1242s | 4.58s/1510s | 118.6s/6221s | 144.2s/9112s |
| avg increase | +19%/-16% | +13%/-11% | +20%/-9% | -11%/-23% | +18%/+1% |

| Aggregators | CIFAR10 | | MolHIV |
|---|---|---|---|
| | Simple | Complex | Simple |
| mean | 83.6s/10526s | 78.7s/10900s | 11.4s/2189s |
| mean $dx_1$ | | | 12.6s/2348s |
| mean $dx_1$ $dx_2$ | 98.4s/8405s | 100.9s/5191s | 14.1s/2345s |
| mean $av_1$ | | | 12.2s/2177s |
| mean $av_1$ $av_2$ | 117.1s/12834s | 89.5s/14481s | 13.9s/2150s |
| mean $dx_1$ $av_1$ | | | 14.0s/2070s |
| avg increase | +29%/+1% | +21%/-10% | +17%/+1% |

## B.4 EIGENVECTOR MULTIPLICITY

The possibility to define equivariant directions using the low-frequency Laplacian eigenvectors is subject to the uniqueness of those vectors. When the dimension of the eigenspaces associated with the lowest eigenvalues is 1, the eigenvectors are defined up to a constant factor. In section 2.4, we propose the use of unit vector normalization and an absolute value to eliminate the scale and sign ambiguity. When the dimension of those eigenspaces is greater than 1, it is not possible to define equivariant directions using the eigenvectors.

Fortunately, it is very rare for the Laplacian matrix to have repeated eigenvalues in real-world datasets. We validate this claim by looking at ZINC and PATTERN datasets where we found no graphs with repeated Fiedler vector and only one graph out of 26k with multiplicity of the second eigenvector greater than 1.

When facing a graph that presents repeated Laplacian eigenvalues, we propose to randomly shuffle, during training time, different eigenvectors randomly sampled in the eigenspace. This technique will act as a data augmentation of the graph during training time allowing the network to train with multiple directions at the same time.

## C    APPENDIX - MATHEMATICAL PROOFS

### C.1    PROOF FOR THEOREM 2.1 (DIRECTIONAL SMOOTHING)

The operation $\boldsymbol{y} = \boldsymbol{B}_{av}\boldsymbol{x}$ is the directional average of $\boldsymbol{x}$, in the sense that $\boldsymbol{y}_u$ is the mean of $\boldsymbol{x}_v$, weighted by the direction and amplitude of $\boldsymbol{F}$.

*Proof.* This should be a simple proof, that if we want a weighted average of our neighbours, we simply need to multiply the weights by each neighbour, and divide by the sum of the weights. Of course, the weights should be positive.

$\square$

### C.2    PROOF FOR THEOREM 2.2 (DIRECTIONAL DERIVATIVE)

Suppose $\hat{\boldsymbol{F}}$ have rows of unit $L^1$ norm. The operation $\boldsymbol{y} = \boldsymbol{B}_{dx}(\hat{\boldsymbol{F}})\boldsymbol{x}$ is the centered directional derivative of $\boldsymbol{x}$ in the direction of $\boldsymbol{F}$, in the sense of equation 4, i.e.

$$\boldsymbol{y} = D_{\hat{\boldsymbol{F}}}\boldsymbol{x} = \left(\hat{\boldsymbol{F}} - \text{diag}\left(\sum_j \hat{\boldsymbol{F}}_{:,j}\right)\right)\boldsymbol{x}$$

*Proof.* Since $\boldsymbol{F}$ rows have unit $L^1$ norm, $\hat{\boldsymbol{F}} = \boldsymbol{F}$. The $i$-th coordinate of the vector $\left(\boldsymbol{F} - \text{diag}\left(\sum_j \boldsymbol{F}_{:,j}\right)\right)\boldsymbol{x}$ is

$$\left(\boldsymbol{F}\boldsymbol{x} - \text{diag}\left(\sum_j \boldsymbol{F}\right)\boldsymbol{x}\right)_i = \sum_j \boldsymbol{F}_{i,j}\boldsymbol{x}(j) - \left(\sum_j \boldsymbol{F}_{i,j}\right)\boldsymbol{x}(i)$$
$$= \sum_{j:(i,j)\in E}(\boldsymbol{x}(j) - \boldsymbol{x}(i))\boldsymbol{F}_{i,j}$$
$$= D_{\boldsymbol{F}}\,\boldsymbol{x}(i)$$

$\square$

### C.3    PROOF FOR THEOREM 2.3 (K-GRADIENT OF THE LOW-FREQUENCY EIGENVECTORS)

Let $\lambda_i$ and $\phi_i$ be the eigenvalues and eigenvectors of the normalized Laplacian of a connected graph $\boldsymbol{L}_{norm}$ and let $a, b = \arg\max_{1\leq i,j\leq n}\{d_K(v_i, v_j)\}$ be the nodes that have highest K-walk distance. Let $m = \arg\min_{1\leq i\leq n}(\phi_1)_i$ and $M = \arg\max_{1\leq i\leq n}(\phi_1)_i$, then $d_K(v_m, v_M) - d_K(v_a, v_b)$ has order $O(1 - \lambda_2)$.

*Proof.* For this theorem, we use the indices $i = 0, ..., (N-1)$, sorted such that $\lambda_i \leq \lambda_{i+1}$. Hence, $\lambda_0 = 0$ and $\lambda_1$ is the first non-trivial eigenvalue.

First we need the following proposition:

**Proposition 1** (K-walk distance matrix). *The K-walk distance matrix $\boldsymbol{P}$ associated with a graph is the matrix such that $(\boldsymbol{P})_{i,j} = d_K(v_i, v_j)$ can be written as $\sum_{p=1}^{K} \boldsymbol{W}^p$, where $\boldsymbol{W} = \boldsymbol{D}^{-1}\boldsymbol{A}$ is the random walk matrix.*

Let's define $\boldsymbol{W} = \boldsymbol{D}^{-1}\boldsymbol{A}$ the random walk matrix of the graph.

First, we are going to show that $W$ is jointly diagonalizable with $\boldsymbol{L}_{\text{norm}}$ and we are going to relate its eigenvectors $\phi_i'$ and its eigenvalues $\lambda_i'$ with the ones of $\boldsymbol{W}$.

Indeed, $\boldsymbol{L}_{\text{sym}}$ is a symmetric real matrix which is semi-positive definite diagonalizable by the spectral theorem. Since the matrix $\boldsymbol{L}_{\text{norm}}$ is similar to $\boldsymbol{D}^{\frac{1}{2}}\boldsymbol{L}_{\text{norm}}\boldsymbol{D}^{-\frac{1}{2}} = \boldsymbol{D}^{-\frac{1}{2}}\boldsymbol{L}\boldsymbol{D}^{-\frac{1}{2}} = \boldsymbol{L}_{\text{sym}}$ and the matrix of similarity is $\boldsymbol{D}^{\frac{1}{2}}$, a positive definite matrix, $\boldsymbol{L}_{\text{norm}}$ is diagonalizable and semi-positive definite.

By

$$\boldsymbol{L}_{\text{norm}} = \boldsymbol{D}^{-1}\boldsymbol{L} = \boldsymbol{D}^{-1}(\boldsymbol{L} + \boldsymbol{D} - \boldsymbol{D}) = \boldsymbol{I} + \boldsymbol{D}^{-1}(\boldsymbol{L} - \boldsymbol{D}) = \boldsymbol{I} - \boldsymbol{D}^{-1}\boldsymbol{A} = \boldsymbol{I} - \boldsymbol{W}$$

the random walk matrix is jointly diagonalizable with the random walk Laplacian. Also their eigenvalues and eigenvectors are related to each other by $\phi_i = \phi_{n-1-i}'$ and $\lambda_i' = 1 - \lambda_{n-1-i}$

Moreover, the constant eigenvector associated with eigenvalue 0 of the Random walk Laplacian, is the eigenvector associated with the highest eigenvalue of the Random walk matrix and by the formula obtained, $\lambda_{n-1}' = 1 - \lambda_0 = 1$

Now, we are going to approximate the K-walk distance matrix $\boldsymbol{P}$ using the 2 eigenvectors of the Random walk matrix associated with the highest eigenvalues.

By Proposition 1 we have that $\boldsymbol{P} = \sum_{p=1}^{K} \boldsymbol{W}^p$, which can be written as

$$\sum_{p=1}^{K}(\sum_{i=0}^{n-1} \phi_i'\phi_i'^T(\lambda_i'))^p = \sum_{p=1}^{K}\sum_{i=0}^{n-1} \phi_i'\phi_i'^T(\lambda_i')^p$$

by eigen-decomposition.

Since $\lambda_{n-1-i} = 1 - \lambda_i'$ and $\lambda_2 \gg \lambda_1$, we have that $\lambda_{n-2}' \gg \lambda_{n-3}'$, hence we can approximate

$$P = \sum_{p=1}^{K}(\sum_{i=0}^{n-1} \phi_i'\phi_i'(\lambda_i')^p) \approx \sum_{p=1}^{K}(\sum_{i=n-2}^{n-1} \phi_i'\phi_i'^T(\lambda_i')^p) + O(\lambda_{n-3}') =$$

$$= \sum_{p=1}^{K}(\sum_{i=0}^{1} \phi_i\phi_i^T(1 - \lambda_i)^p) + O(1 - \lambda_2) = \sum_{p=1}^{K}(\phi_0\phi_0^T + \phi_1\phi_1^T(1 - \lambda_1)^p) + O(1 - \lambda_2)$$

$$= K\phi_0\phi_0^T + \kappa\phi_1\phi_1^T + O(1 - \lambda_2)$$

where $\kappa = \sum_{p=1}^{K}(1 - \lambda_1)^p$ is a positive constant.

Now we are going to show that the farthest nodes with respect to the $K$-walk distance are the ones associated with the highest and lowest value of $\phi_1$.

Indeed if we want to choose $i, j$ to be at the farthest distance we need to minimise

$$(\boldsymbol{P})_{i,j} = (K\phi_0\phi_0^T + \kappa\phi_1\phi_1^T)_{i,j} = \frac{K}{n} + \kappa\phi_1(i)\phi_1(j)$$

which is minimum when $\phi_1(i)\phi_1(j)$ is minimum.

We are going to show that exist $p, q$ such that $\phi_1(p) < 0, \phi_1(q) > 0$. Since the eigenvector is non-zero, without loss of generality assume $\phi_1(0) \neq 0$. Since $\phi_0$ and $\phi_1$ are eigenvectors associated with different eigenvalues of a real symmetric matrix, they are orthogonal:

$$\sum_{i=0}^{n-1} \phi_0(i) \cdot \phi_1(i) = 0$$

and since $\phi_0$ is constant the previous equation leads to

$$\sum_{i=0}^{n-1} \phi_1(i) = 0 \iff \phi_1(0) = -\sum_{i=1}^{n-1} \phi_1(i)$$

If such $p, q$ didn't exist then we would get that $\forall i, j \; \phi_1(i) \cdot \phi_1(j) \geq 0$, hence multiplying both sides of the previous equation by $\phi_1(0)$ we get

$$\phi_1(0)^2 = -\sum_{i=1}^{n-1} \phi_1(i) \cdot \phi_1(0) \Rightarrow \phi_1(0)^2 \leq 0$$

Which is a contradiction since by assumption $\phi_1(0) > 0$; hence exist $p, q$ such that $\phi_1(p) < 0, \phi_1(q) > 0$.

Since $\phi_1$ attains both positive and negative values, the quantity $\phi_1(i)\phi_1(j)$ is minimised when it has negative sign and highest absolute value, hence when $i, j$ are associated with the negative and positive values with the highest absolute value: the lowest and the highest value of $\phi_1$. Hence, $d_K(v_M, v_m) - d_K(v_a, v_b) = O(1 - \lambda_2)$

$\square$

### C.4    Informal argument in support of Conjecture 2.4 (Gradient steps reduce expected hitting time)

Suppose that $x, y$ are uniformly distributed random nodes such that $\phi_i(x) < \phi_i(y)$. Let $z$ be the node obtained from $x$ by taking one step in the direction of $\nabla \phi_i$, then the expected hitting time is decreased proportionally to $\lambda_i^{-1}$ and

$$\mathbb{E}_{x,y}[Q(z, y)] \leq \mathbb{E}_{x,y}[Q(x, y)]$$

As a reminder, the definition of a gradient step is given in the definition 3, copied below.

Suppose the two neighboring nodes $x$ and $z$ are such that $\phi(z) - \phi(x)$ is maximal among the neighbors of $x$, then we will say $z$ is obtained from $x$ by taking a step in the direction of the gradient $\nabla \phi$.

In (Chung & S.T.Yau, 2000), it is shown the hitting time $Q(x, y)$ is given by the equation

$$Q(x, y) = vol \left( \frac{\boldsymbol{G}(y, y)}{d_y} - \frac{\boldsymbol{G}(x, y)}{d_x} \right)$$

With $\lambda_k$ and $\phi_k$ being the $k$-th eigenvalues and eigenvectors of the symmetric normalized Laplacian $L_{\text{sym}}$, $vol$ the sum of the degrees of all nodes, $d_x$ the degree of node $x$ and $\boldsymbol{G}$ Green's function for the graph

$$\boldsymbol{G}(x, y) = d_x^{\frac{1}{2}} d_y^{\frac{-1}{2}} \sum_{k>0} \frac{1}{\lambda_k} \phi_k(x) \phi_k(y)$$

Since the sign of the eigenvector is not deterministic, the choice $\phi_i(x) < \phi_i(y)$ is used to simplify the argument without having to consider the change in sign.

Supposing $\lambda_1 \ll \lambda_2$, the first term of the sum of $\boldsymbol{G}$ has much more weight than the following terms. With $z$ obtained from $x$ by taking a step in the direction of the gradient of $\phi_1$ we have

$$\phi_1(z) - \phi_1(x) > 0$$

We want to show that the following inequality holds

$$\mathbb{E}_{x,y}(Q(z, y)) < \mathbb{E}_{x,y}(Q(x, y))$$

this is equivalent to the following inequality

$$\mathbb{E}_{x,y}[\boldsymbol{G}(z, y)] > \mathbb{E}_{x,y}[\boldsymbol{G}(x, y)]$$

By the hypothesis $\lambda_1 \ll \lambda_2$, we can approximate $\boldsymbol{G}(x,y) \sim d_x^{\frac{1}{2}} d_y^{\frac{-1}{2}} \frac{1}{\lambda_1} \phi_1(x)\phi_1(y)$ so the last inequality is equivalent to

$$\mathbb{E}_{x,y}\left[d_z^{\frac{1}{2}} d_y^{\frac{-1}{2}} \frac{1}{\lambda_1}\phi_1(z)\phi_1(y)\right] > \mathbb{E}_{x,y}\left[d_x^{\frac{1}{2}} d_y^{\frac{-1}{2}} \frac{1}{\lambda_1}\phi_1(x)\phi_1(y)\right]$$

Removing all equal terms from both sides, the inequality is equivalent to

$$\mathbb{E}_{x,y}\left[d_z^{\frac{1}{2}}\phi_1(z)\right] > \mathbb{E}_{x,y}\left[d_x^{\frac{1}{2}}\phi_1(x)\right]$$

But showing this last inequality is not easy. We know that $\phi_1(z) > \phi_1(x)$ and from the choice of $z$ being a step in the direction of $\nabla\phi_1$, we know it is less likely to be on the border of the graph so we believe $\mathbb{E}(d_z) \geq \mathbb{E}(d_x)$. Thus we also believe that the conjecture should hold in general.

We believe this should be true even without the assumption on $\lambda_1$ and $\lambda_2$ and for more eigenvectors than $\phi_1$.

## C.5 Proof for Lemma C.1 (Cosine eigenvectors)

Consider the lattice graph $\Gamma$ of size $N_1 \times N_2 \times ... \times N_n$, that has vertices $\prod_{i=1,...,n}\{1, ..., N_i\}$ and the vertices $(x_i)_{i=1,...,n}$ and $(y_i)_{i=1,...,n}$ are connected by an edge iff $|x_i - y_i| = 1$ for one index $i$ and $0$ for all other indices. Note that there are no diagonal edges in the lattice. The eigenvector of the Laplacian of the grid $L(\Gamma)$ are given by $\phi_j$.

**Lemma C.1** (Cosine eigenvectors). *The Laplacian of $\Gamma$ has an eigenvalue $2 - 2\cos\left(\frac{\pi}{N_i}\right)$ with the associated eigenvector $\phi_j$ that depends only the variable in the $i$-th dimension and is constant in all others, with $\phi_j = \mathbf{1}_{N_1} \otimes \mathbf{1}_{N_2} \otimes ... \otimes \boldsymbol{x}_{1,N_i} \otimes ... \otimes \mathbf{1}_{N_n}$, and $\boldsymbol{x}_{1,N_i}(j) = \cos\left(\frac{\pi j}{n} - \frac{\pi}{2n}\right)$*

*Proof.* First, recall the well known result that the path graph on $N$ vertices $P_N$ has eigenvalues

$$\lambda_k = 2 - 2\cos\left(\frac{\pi k}{n}\right)$$

with associated eigenvector $\boldsymbol{x}_k$ with $i$-th coordinate

$$\boldsymbol{x}_k(i) = \cos\left(\frac{\pi k i}{n} + \frac{\pi k}{2n}\right)$$

The Cartesian product of two graphs $G = (V_G, E_G)$ and $H = (V_H, E_H)$ is defined as $G \times H = (V_{G\times H}, E_{G\times H})$ with $V_{G\times H} = V_G \times V_H$ and $((u_1, u_2), ((v_1, v_2)) \in E_{G\times H}$ iff either $u_1 = v_1$ and $(u_2, v_2) \in E_H$ or $(u_1, v_1) \in V_G$ and $u_2 = v_2$. It is shown in (Fiedler, 1973) that if $(\mu_i)_{i=1,...,m}$ and $(\lambda_j)_{j=1,...,n}$ are the eigenvalues of $G$ and $H$ respectively, then the eigenvalues of the Cartesian product graph $G \times H$ are $\mu_i + \lambda_j$ for all possible eigenvalues $\mu_i$ and $\lambda_j$. Also, the eigenvectors associated to the eigenvalue $\mu_i + \lambda_j$ are $u_i \otimes v_j$ with $u_i$ an eigenvector of the Laplacian of $G$ associated to the eigenvalue $\mu_i$ and $v_j$ an eigenvector of the Laplacian of $H$ associated to the eigenvalue $\lambda_j$.

Finally, noticing that a lattice of shape $N_1 \times N_2 \times ... \times N_n$ is really the Cartesian product of path graphs of length $N_1$ up to $N_n$, we conclude that there are eigenvalues $2 - 2\cos\left(\frac{\pi}{N_i}\right)$. Denoting by $\mathbf{1}_{N_j}$ the vector in $\boldsymbol{R}^{N_j}$ with only ones as coordinates, then the eigenvector associated to the eigenvalue $2 - 2\cos\left(\frac{\pi}{N_i}\right)$ is

$$\mathbf{1}_{N_1} \otimes \mathbf{1}_{N_2} \otimes ... \otimes \boldsymbol{x}_{1,N_i} \otimes ... \otimes \mathbf{1}_{N_n}$$

where $\boldsymbol{x}_{1,N_i}$ is the eigenvector of the Laplacian of $P_{N_i}$ associated to its first non-zero eigenvalue. $2 - 2\cos\left(\frac{\pi}{N_i}\right)$. $\qquad\square$

### C.6 Radius 1 convolution kernels in a grid

In this section we show any radius 1 convolution kernel can be obtained as a linear combination of the $\boldsymbol{B}_{dx}(\nabla\phi_i)$ and $\boldsymbol{B}_{av}(\nabla\phi_i)$ matrices for the right choice of Laplacian eigenvectors $\phi_i$. First we show this can be done for 1-d convolution kernels.

**Theorem C.2.** *On a path graph, any 1D convolution kernel of size 3 $k$ is a linear combination of the aggregators $\boldsymbol{B}_{av}, \boldsymbol{B}_{dx}$ and the identity $\boldsymbol{I}$.*

*Proof.* Recall from the previous proof that the first non zero eigenvalue of the path graph $P_N$ has associated eigenvector $\phi_1(i) = \cos(\frac{\pi i}{N} - \frac{\pi}{2N})$. Since this is a monotone decreasing function in $i$, the $i$-th row of $\nabla\phi_1$ will be

$$(0, ..., 0, s_{i-1}, 0, -s_{i+1}, 0, ..., 0)$$

with $s_{i-1}$ and $s_{i+1} > 0$. We are trying to solve

$$(a\boldsymbol{B}_{av} + b\boldsymbol{B}_{dx} + c\mathbf{Id})_{i,:} = (0, ..., 0, x, y, z, 0, ..., 0)$$

with $x, y, z$, in positions $i - 1, i$ and $i + 1$. This simplifies to solving

$$a\frac{1}{\|s\|_{L^1}}|s| + b\frac{1}{\|s\|_{L^2}}s + c(0, 1, 0) = (x, y, z)$$

with $s = (s_{i-1}, 0, -s_{i+1})$, which always has a solution because $s_{i-1}, s_{i+1} > 0$. $\square$

**Theorem C.3** (Generalization radius-1 convolutional kernel in a grid). *Let $\Gamma$ be the $n$-dimensional lattice as above and let $\phi_j$ be the eigenvectors of the Laplacian of the lattice as in theorem C.1. Then any radius 1 kernel $k$ on $\Gamma$ is a linear combination of the aggregators $\boldsymbol{B}_{av}(\phi_i), \boldsymbol{B}_{dx}(\phi_i)$ and $\boldsymbol{I}$.*

*Proof.* This is a direct consequence of C.2 obtained by adding $n$ 1-dimensional kernels, with each kernel being in a different axis of the grid as per Lemma C.1. See figure 4 for a visual example in 2D.

$\square$

### C.7 Proof for Theorem 2.7 (Generalization radius-$R$ convolutional kernel in a lattice)

For an $n$-dimensional lattice, any convolutional kernel of radius $R$ can be realized by a linear combination of directional aggregation matrices and their compositions.

*Proof.* For clarity, we first do the 2 dimensional case for a radius 2, then extended to the general case. Let $k$ be the radius 2 kernel on a grid represented by the matrix

$$\boldsymbol{a}_{5\times5} = \begin{pmatrix} 0 & 0 & a_{-2,0} & 0 & 0 \\ 0 & a_{-1,-1} & a_{-1,0} & a_{-1,1} & 0 \\ a_{0,-2} & a_{0,-1} & a_{0,0} & a_{0,1} & a_{0,2} \\ 0 & a_{1,-1} & a_{1,0} & a_{1,1} & 0 \\ 0 & 0 & a_{2,0} & 0 & 0 \end{pmatrix}$$

since we supposed the $N_1 \times N_2$ grid was such that $N_1 > N_2$, by theorem C.1, we have that $\phi_1$ is depending only in the first variable $x_1$ and is monotone in $x_1$. Recall from C.1 that

$$\phi_1(i) = \cos\left(\frac{\pi i}{N_1} + \frac{\pi}{2N_1}\right)$$

The vector $\frac{N_1}{\pi}\nabla\arccos(\phi_1)$ will be denoted by $\boldsymbol{F}_1$ in the rest. Notice all entries of $\boldsymbol{F}_1$ are 0 or $\pm 1$. Denote by $\boldsymbol{F}_2$ the gradient vector $\frac{N_2}{\pi}\nabla\arccos(\phi_k)$ where $\phi_k$ is the eigenvector given by theorem C.1 that is depending only in the second variable $x_2$ and is monotone in $x_1$ and recall

$$\phi_k(i) = \cos\left(\frac{\pi i}{N_2} + \frac{\pi}{2N_2}\right)$$

For a matrix $\boldsymbol{B}$, let $\boldsymbol{B}^{\pm}$ the positive/negative parts of $\boldsymbol{B}$, ie matrices with positive entries such that $\boldsymbol{B} = \boldsymbol{B}^+ - \boldsymbol{B}^-$. Let $\boldsymbol{B}_{r1}$ be a matrix representing the radius 1 kernel with weights

$$
\boldsymbol{a}_{3\times3} = \begin{pmatrix} 0 & a_{-1,0} & 0 \\ a_{0,-1} & a_{0,0} & a_{0,1} \\ 0 & a_{1,0} & 0 \end{pmatrix}
$$

The matrix $\boldsymbol{B}_{r1}$ can be obtained by theorem C.3. Then the radius 2 kernel $k$ is defined by all the possible combinations of 2 positive/negative steps, plus the initial radius-1 kernel.

$$
\boldsymbol{B}_{r2} = \underbrace{\sum_{\substack{-2 \leq i,j \leq 2 \\ |i|+|j|=2}} \left( a_{i,j} (\boldsymbol{F}_1^{sgn(i)})^{|i|} (\boldsymbol{F}_2^{sgn(j)})^{|j|} \right)}_{\text{Any combination of 2 steps}} + \underbrace{\boldsymbol{B}_{r1}}_{\text{all possible single-steps}}
$$

with $sgn$ the sign function $sgn(i) = +$ if $i \geq 0$ and $-$ if $i < 0$. The matrix $\boldsymbol{B}_{r2}$ then realises the kernel $\boldsymbol{a}_{5\times5}$.

We can further extend the above construction to $N$ dimension grids and radius $R$ kernels $k$

$$
\underbrace{\sum_{\substack{V=\{v_1,v_2,...,v_N\}\in\mathbb{N}^n \\ ||V||_{L^1}\leq R \\ -R\leq v_i\leq R}}}_{\text{Any choice of walk } V \text{ with at most } R\text{-steps}} a_V \underbrace{\prod_{j=1}^{N} (\boldsymbol{F}_j^{sgn(v_j)})^{|v_j|}}_{\text{Aggregator following the steps defined in } V}
$$

with $\boldsymbol{F}_j = \frac{N_j}{\pi} \nabla \arccos \boldsymbol{\phi}_j$ , $\boldsymbol{\phi}_j$ the eigenvector with lowest eigenvalue only dependent on the $j$-th variable and given in theorem C.1 and $\prod$ is the matrix multiplication. $V$ represents all the choices of walk $\{v_1, v_2, ..., v_n\}$ in the direction of the fields $\{\boldsymbol{F}_1, \boldsymbol{F}_2, ..., \boldsymbol{F}_n\}$. For example, $V = \{3, 1, 0, -2\}$ has a radius $R = 6$, with 3 steps *forward* of $\boldsymbol{F}_1$, 1 step *forward* of $\boldsymbol{F}_2$, and 2 steps *backward* of $\boldsymbol{F}_4$.

$\square$

## C.8 Proof for Theorem 2.8 (Comparison with 1-WL test)

DGNs using the *mean* aggregator, any directional aggregator of the first eigenvector and injective degree-scalers are strictly more powerful than the 1-WL test.

*Proof.* We will show that (1) DGNs are at least as powerful as the 1-WL test and (2) there is a pair of graphs which are not distinguishable by the 1-WL test which DGNs can discriminate.

Since the DGNs include the mean aggregator combined with at least an injective degree-scaler, Corso et al. (2020) show that the resulting architecture is at least as powerful as the 1-WL test.

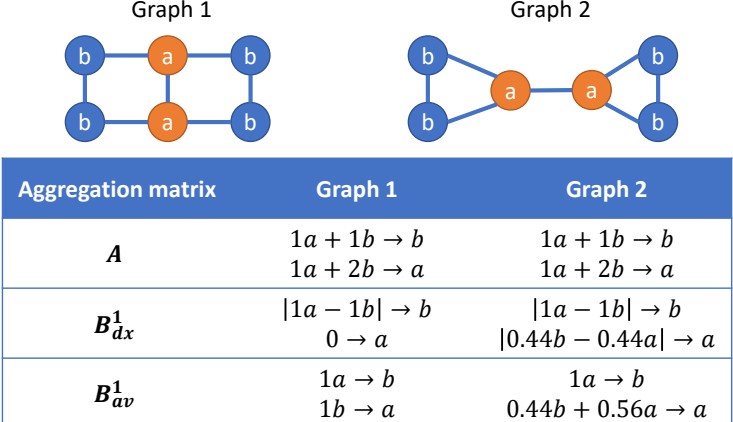

| Aggregation matrix | Graph 1 | Graph 2 |
|:---:|:---:|:---:|
| $A$ | $1a + 1b \rightarrow b$ 
 $1a + 2b \rightarrow a$ | $1a + 1b \rightarrow b$ 
 $1a + 2b \rightarrow a$ |
| $B_{dx}^1$ | $\|1a - 1b\| \rightarrow b$ 
 $0 \rightarrow a$ | $\|1a - 1b\| \rightarrow b$ 
 $\|0.44b - 0.44a\| \rightarrow a$ |
| $B_{av}^1$ | $1a \rightarrow b$ 
 $1b \rightarrow a$ | $1a \rightarrow b$ 
 $0.44b + 0.56a \rightarrow a$ |

Figure 8: Illustration of an example pair of graphs which the 1-WL test cannot distinguish but DGNs can. The table shows the node feature updates done at every layer. MPNN with mean/sum aggregators and the 1-WL test only use the updates in the first row and therefore cannot distinguish between the nodes in the two graphs. DGNs also use directional aggregators that, with the vector field given by the first eigenvector of the Laplacian matrix, provides different updates to the nodes in the two graphs.

Then, to show that the DGNs are strictly more powerful than the 1-WL test it suffices to provide an example of a pair of graphs that DGNs can differentiate and 1-WL cannot. Such a pair of graphs is illustrated in figure 8.

The 1-WL test (as any MPNN with, for example, sum aggregator) will always have the same features for all the nodes labelled with $a$ and for all the nodes labelled with $b$ and, therefore, will classify the graphs as isomorphic. DGNs, via the directional smoothing or directional derivative aggregators based on the first eigenvector of the Laplacian matrix, will update the features of the $a$ nodes differently in the two graphs (figure 8 presents also the aggregation functions) and will, therefore, be capable of distinguishing them.

$\square$

