# OpenReview forum: "Directional graph networks"
_ICLR.cc/2021/Conference — Reject_

### Official Review · AnonReviewer1 · 2020-10-27
**Review Comment #1**

**Rating:** 5
**Confidence:** 3

**Review:**

**Revision History**
- Oct. 27th 2020: Submit the initial comment.
- Nov. 20th, 2020: Fix the "Correctness" part for better readability.


**Review summary**

The use of vector fields on a graph to define directed convolution is an interesting idea. In addition, the proposed GNN is theoretically sound in that it is an extension of CNNs on a grid graph in a certain sense (Theorem 2.7). Empirically, the proposed GNN performs better than existing methods. Nevertheless, I am not confident that the proposed method effectively reduces the over-smoothing and over-squashing problems theoretically or empirically. Regarding the data augmentation, although the proposed augmentation method works empirically, I am not sure the soundness of the method.

**Summary of the paper**

This paper proposed DGN (directional graph neural network), which employs directional aggregations using a vector field on a graph. Specifically, it proposed two types of aggregations: Directional averaging and Directional derivative matrix, which use eigenvectors associated with a graph Laplacian. DGN generalized a CNN when the underlying graph is a grid graph. Also, it proposed a data augmentation method using vector fields. This paper claimed that the proposed methods reduce the over-smoothing and over-squashing problem and perform well compared to existing methods.

**Claim**

If I understand correctly, the main claims of this paper are as follows. I assume them and evaluate the paper based on whether it supports them.
- Claim 1: The lack of propagation directions in existing GNNs limits the discriminative power of GNNs (Section 2.1, Paragraph 2)
- Claim 2: Existing GNNs that used asymmetric graph kernels do not use the graph topology and directional flow (Section 1, Paragraph 2).
- Claim 3: DGN generalizes CNNs on a grid (Section 1, Paragraph 4) and existing aggregation (Section A.1).
- Claim 4: DGN works as a countermeasure of over-smoothing and over-squashing problems (Section 2 Corollary2.5, Corollary 2.6). In particular, DGN theoretically and empirically allows message passing across distance layers (Section 1, Paragraph 5).
- Claim 5: Proposed data augmentation method using a vector field on a graph is effective.

**Soundness of the claims**

Can theory support the claims?
- Regarding Claim 1 and 2, this paper did not compare (at least directly) with existing methods theoretically.
- Claim 3 is supported by Theorem 2.7.
- Claim 4 is supported by Corollary 2.5, 2.6. (But I am not entirely sure their correctness. See "Correctness" section.)
- Claim 5 is not theoretically supported. But I am uncertain whether this paper argues that this claim has theoretical justifications.

Can empirical evaluation support the claims?
- Claim 1 is empirically supported by comparing GCN, GraphSage, GIN, and MoNet in the experiment in Figure 5.
- Claim 2 is empirically supported by the comparison with GAT and GatedGCN in the experiment in Figure 5.
- Claim 3 is theoretical. So, it is OK that no experiments correspond to it.
- Claim 4 is validated indirectly by comparing the proposed model with existing ones known to be suffered from the over-smoothing and over-squashing problem. If I do not miss any information, this paper did not compare the proposed method with anti-over-smoothing methods such as DropEdge [Rong et al., 20].
- Claim 5 seems to have some support from the empirical assessment (Figure 6). But I am not perfectly certain of their correctness. (See "Correctness" section.)

[Rong et al., 20] Rong, Y., Huang, W., Xu, T., & Huang, J. DropEdge: Towards deep graph convolutional networks on node classification. ICLR 2020.

**Significance and novelty**

Novelty
- The idea of using a vector field for directional message passing is novel, to the best of my knowledge.

Relation to previous work
- The authors mentioned existing GNNs that employed asymmetric kernels (such as GAT or DimeNet). Although they discussed the difference of the proposed model from them in detail, I think is these models are based on different design principles. (of course, it would be great if this paper discusses the difference in more details)
- As I wrote previously, this paper did not compare the proposed method with existing anti-over-smoothing methods.

**Correctness**

Is the theory correct?

- Theorem 2.3: I think the statement of Theorem2.3 is informal. I would recommend to write the formal statement somewhere, at least in the appendix.
- Section 2.4: This paper wrote that "These results from theorem 2.3 and conjecture 2.4 have the following immediate corollaries" (the sentence after Conjecture 2.4). This sentence is not appropriate, as it reads that this paper derives corollaries from a conjecture, which is not proven yet. Considering that, it looks that Corollary 2.4 and 2.5 are not proven theoretically.
- Conjecture 2.4: I could not understand what "one step direction of $\nabla \phi_i$" meant.  I understand it when a vector field in a Euclidean space. However, since $\nabla \phi_i$ is a function on the set of edges on a graph, this analogy does not hold. I want this paper to clarify it.
- Theorem 2.7: The proof seems correct. The idea is to pick eigenvectors $\phi_1, \ldots, \phi_D$ that only change along each axis and take vector fields $\nabla \arccos(\phi_d)$ as basic components.
- Definition 4: Regarding the data augmentation (Definition 4), I understand the intuition that we "rotate" a vector field in the reverse direction instead of rotating an image. However, I could not understand whether the definition is sound.  Specifically, if I understand correctly, I expect that $F'_1 = F_1$ and $F'_2=F_2$ when $\theta = 0$. However, the latter one does not hold in general. Let us decompose $F_2$ into directions that is parallel and perpendicular to $F_1$ as $F_2 = F_2^{\parallel} + kF_2^{\perp}$ ($k=\|F_2-F_2^\parallel\|$). Then, it holds $F_2'|_{\theta=0}=F_1\cos \alpha+F_2^\perp\sin \alpha=F_2^{\parallel}+F_2^\perp\sin \alpha$. This does not equal $F_2$ in general.

Is the experimental evaluation correct?
- In Section 2.7, this paper proposed the random distortion of a vector field as Definition 5. However, if I do not miss anything, it only defined the concept and did not discuss its theoretical nor empirical properties. Therefore, I do not think the proposed augmentation method is justified.
- In Figure 6, when the maximum rotation angle is $45^\circ$,, the training and test accuracy of the dx method is worse than the baseline. I want this paper to discuss the behavior.

Is the experiment reproducible?
- Yes

**Clarity**

Can I understand the main point of the paper easily?
- Yes. The background of the proposed method is explained in Section 2.2. The proposed method is explained in Section 2.3. Proofs are easy to understand. The explanations are easy to understand.

Is the organization of paper well?
- Yes. I did not find any problem regarding the organization of the paper.

Are figures and tables appropriately made?
- Yes

**Additional feedback**

- Section 2.3, Figure 2: Should we replace $u_i$ with $x_{u_i}$, as $u_i$ is a label for the node itself?
- Section 2.4, Theorem 2.3: There seems to be inconsistency of ranges the index $i$ runs in different places. Specifically, they assume that indices of $\lambda_i$ and $\lambda_i'$ run through $i=0, ..., n-1$ in one place and $i=1, ..., n$ in another place. Also, it should be explicitly written that $\lambda_i$'s are sorted (either in ascending or descending order) because otherwise $\lambda_1$ and $\lambda_2$ are undefined.
- Section 2.5, page 5, last line: figure 2.5 → figure 3
- Section 3, page 7, paragraph 3: How can we determine $k$?
- Section 4, page 7, paragraph 1: What does $i$ in $B^i_{dx}$ and $B^i_{av}$ indicate? (index of the eigenvector?）
- Section C.3, page 15, Make $D^{-1/2}$ bold
- Figure 4 referred to $B_{av}$ as av, while Figure 5 referred as smooth. I would recommend to make them consistent.
- In Equation (8), they normalized the length of the vector field to 1 in the definition of the perpendicular component of a vector field. However, I think we usually refer to the unnormalized vector field (i.e., denominator of (8)) as a perpendicular component.

---

> ### Author Response · Authors · 2020-11-18
> **Answer to reviewer #1**
>
> We would like to thank you for the time spent to give such a thorough and detailed review of our work. We find your comments very constructive and will try to answer your concerns below.
>
> ## Summary
> We improved the clarity of the paper by adding Figure 1 to summarize the steps, equations and architecture of the proposed DGN model. We improved the theoretical framework by proving that the proposed aggregation is more powerful than a 1-WL test, thus more expressive than many standard GNNs. The notation was improved in the “Data augmentation” section, where we also fixed the error for the rotation equation, and demonstrated empirically the validity of the distortion augmentation with new test results
>
> ## Claim
> **Claim 5** The claim is not only that the data augmentation is effective, but also that some of the proposed rotation and reflection are not possible with standard GNNs, which highlights the versatility of the proposed vector fields
>
> ## Soundness of the claims
> **Can theory support the claims?**
>
> **Claim 1** We added a demonstration that the method is strictly more powerful than the 1-WL (Weisfeiler-Lehman) test in section 2.7, theorem 2.8 and appendix C.8. Other GNNs (e.g. GCN, GIN, GAT, MPNN, PNA) are not more powerful than the 1-WL test since they either use mean/sum/max aggregation, or an attention between neighbouring nodes.
>
> **Claim 2** We believe that a theoretical argument is not needed since it is known from their architecture that other GNN do not embed the global topology of the graph in their architecture
>
> **Claim 5** We provide no theoretical justification for the data augmentation, but we proved that DGNs are a generalization of CNNs and such augmentation have been very successful in Computer Vision. Further, we think that graph datasets will benefit from it since they are often small, thus motivating the proposed augmentation
>
> **Can empirical evaluation support the claims?**
>
> **Claim 4** Thank you for pointing out the DropEdge paper, we added it in our literature review.
> However, we do not think that our method competes with DropEdge since the concept is very different. In fact, both methods are complementary rather than competitive since DropEdge can be added to our DGN model, either for the mean or for the directional aggregations. Moreover, we do not consider our DGN model to be an anti-over-smoothing method, rather it is a novel GNN layer, with one of it’s advantages being to have less over-smoothing than conventional layers.
>
> ## Correctness
> **Is the theory correct?**
>
> We formalized theorem 2.3 by adding a boundary to the approximation that the extremities of the eigenvector are the nodes with the longest K-Walk distance.
> Since both corollaries can be derived from theorem 2.3, we changed the wording to “The next two corollaries follow from theorem 2.3 (and also conjecture 2.4 if it is  true).”
> We clarified the meaning of “one step in the direction of $\nabla \phi_i$” by adding the definition 3 to explain that the step from $x$ to its neighbour $z$ is done to maximize $\phi(z) - \phi (x)$.
>
> We thank you for noticing the error in the definition of the vector rotation. The equations were developed for vectors of unit L2-norm (in which case they work) but do not generalize to all vectors. We improved the general notation and modified the text to make sure that this L2-norm condition is clear. We now define the fields $\hat{F}$ to emphasize that they are different from $F$ by being row-normalized. This error does not affect our empirical results since our implementation already used unit-norm vectors.
>
> **Is the experimental evaluation correct?**
>
> To answer your concern, we added empirical results with vector field distortion in Fig 7, and demonstrate that it improves the performance of the derivative aggregator with a similar behavior to the rotation augmentation
>
> We added a few sentences to the “Data augmentation” paragraph of section 4 to explain why the performance is lower than the baseline when there is a rotation of 45 degrees or a distortion of 40%
>
> *As expected, the performance decreases when the rotation or distortion is too high since the augmented graph changes too much. In computer vision, images similar to CIFAR10 are usually rotated by less than $30^\circ$ [1,2]. Further, due to the constant number of parameters across models, less parameters are attributed to the mean aggregation in the directional models, thus it cannot fit well the data when the rotation/distortion is too strong since the directions become less informative. We expect larger models to perform better at high angles.*
>
> ## Additional feedback
>
> We answered all additional feedback in the paper, fixing every error that was pointed out and clarifying the ambiguous notations.
>
> We thank you again for your thorough review and hope to have answered your concerns. Please let us know if you have any remaining questions or concerns regarding our work.
>
> [1] doi.org/10.1186/s40537-019-0197-0
>
> [2] doi.org/10.21427/148b-ar75

---

> > ### Comment · AnonReviewer1 · 2020-11-20
> > **Reply to authors' comment**
> >
> > First of all, I thank the authors for taking my comment into account seriously and updating the manuscript. Here is the initial reply to the authors' comments. I will take some time to check the updated theorems and give the authors questions and comments if necessary.
> >
> > ## Claim
> >
> > **Claim 5**: The proposed vector-field method enabled us to define the rotation and reflection. I understand that the authors claimed the definition itself is novel since it is impossible for standard GNNs (correct me if I am wrong).
> >
> > ## Soundness of the claims
> >
> > **Claim 1**: I shall check Theorem 2.8.
> >
> > **Claim 4**: I understand that DropEdge is not a competitive method but a complementary one. Regarding the term "anti-over-smoothing," I am sorry that I used it without definition. I intended that both DropEdge and the proposed method are "anti-over-smoothing" methods. In other words, I imagined not only plug-in methods (i.e., a method to reduce the over-smoothing problem of the original GNN) but also components of GNNs (i.e., a GNN layer which does not have the over-smoothening problem).
> >
> > ## Correctness
> >
> > **Theorem 2.3**: Thank you for reflecting on my comment. Let me take time to read the proof, along with the connection with Corollary 2.5 and 2.6. For now, I have one comment: I think the proof implicitly used the fact that $\phi_1$ has at least both positive and negative components. We can prove it as follows. Since eigenvectors associated with different eigenvalues are orthogonal, we have $\langle \phi_0, \phi_1 \rangle = 0$. Besides, $\phi_0$ is the half-power of degree vector, consisting of positive values. Combining the two above, we can prove the fact.
> >
> >
> > **Definition 5**: Thank you for updating the definition. I think the formula themselves are consistent. However, in my understanding, the perpendicular component of $\hat{F}_2$ is usually defined as $\hat{F}_2^\perp := \hat{F}_2 - \langle \hat{F}_1, \hat{F}_2\rangle\hat{F}_1$ (i.e., without normalization) so that the projection of $\hat{F}_2$ onto the hyperplane perpendicular to $\hat{F}_1$ is the same as $\hat{F}_2^\perp$.

---

> > > ### Author Response · Authors · 2020-11-24
> > > **Second reply to Reviewer 1**
> > >
> > > We would like to thank you again for the meticulous revision of the paper.
> > >
> > > Regarding Theorem 2.3 you are correct. We were implicitly assuming the Laplacian eigenvectors to have both positive and negative components. As you suggested, we have added the proof of this claim.
> > >
> > > Regarding Definition 5, we added the word **normalized** in the definition of $\hat{F}^\perp_2$ given below
> > >
> > > *The vector field $\hat{F}^\perp_2$ is the normalized component of $\hat{F}_2$ perpendicular to $\hat{F}_1$.*
> > >
> > > Considering the novelty of the paper and the numerous theoretical developments required to support the claims, we are very thankful of the time you spent reviewing each claim and theorem individually, as it will ensure the correctness of the work and increase it's potential impact. We hope that our newest version satisfies most of your concerns.

---

### Official Review · AnonReviewer3 · 2020-10-29
**Great novel graph method which brings graph nets closer to CNN - but questions about equivariance**

**Rating:** 7
**Confidence:** 4

**Review:**

Summary:
The authors propose a convolution as a message passing of node features over edges where messages are aggregated weighted by a "direction" edge field. Furthermore, the authors propose to use the gradients of Laplace eigenfunctions as direction fields. Presumably, the aggregation is done with different direction fields derived from the Laplace eigenfunctions with lowest eigenvalues, which are then linearly combined with learnable parameters. Doing so allows their graph network to behave more like a conventional CNN, in which the kernels have different parameters for signals from different directions. The authors achieve good results on several benchmarks. Furthermore, the authors prove that their method reduces to a conventional CNN on a rectangular grid and have theoretical results that suggest that their method suffers less from the "over-smoothing" and "over-squashing" problems.

Strong points:
The proposal is highly novel. It is a simple and scalable modification to conventional graph nets that shows a strong performance increase in the benchmarks. The paper is mostly clearly written. The theoretical analyses contribute to the understanding of the work.

Weak points:
- It is unclear how parameters are used in their model. I presume that they apply the directional derivative with several edge fields and then linearly combine with learnable parameters, but this is not stated explicitly. Furthermore, for different graphs with different spectra, how are the parameter shared? Simply by their order?
- As the authors note at the end of Sec 2.4, the Laplace eigenvectors can not uniquely be identified, only the eigenspace of a certain eigenvalue. For example, this means that when Lanczos’ method is applied to two isomorphic graphs in different orderings, different eigenvectors can be returned. The authors propose to overcome the arbitrary-ness of the sign by taking the absolute value after each dictional derivative. Still, when an eigenvector is used from a degenerate eigenspace, the method appears not equivariant to node re-orderings. The authors state that in their datasets, the first non-trivial eigenvector is always non-degenerate, but the experiments also use higher eigenvectors. Are these also non-degenerate? Or is that model not equivariant? This also means that when using the proposed method on a square grid, which the authors do on CIFAR10, the method is not equivariant to node re-orderings.

Recommendation:
I recommend to accept this paper, as it proposes a simple to use method to build more powerful graph nets. In spite of my concerns about equivariance, it appears to perform well in relevant benchmarks.

Opportunities for improvement:
- The paper could be improved by more directly addressing the concerns about equivariance. Are some of the proposed models indeed not equivariant on certain graphs? Do we care that the model is not always equivariant?
- It would be interesting to see if the authors could prove that the resulting model is more expressive than a conventional graph networks, for example by comparing theoretically or experimentally to the expressiveness of Weisfeiler-Lehman tests.

### Post rebuttal
My previous rating still applies. If accepted, I encourage the authors to more clearly state in the final version that their method is not applicable (without additional - and arguably inelegant - random augmentation) to graphs with degenerate eigenvalues and in particular symmetric graphs. The necessity of taking the absolute value to ensure invariance to the sign of the eigenvector should also be more clearly stated.
I share reviewer #1's concerns about corollaries 2.5 and 2.6. These should be clarified or removed from the final version. Nevertheless, I think the novelty of the approach justifies acceptance. It is a simple modification that may bring the expressive power of graph networks closer to that of pixel CNNs.

---

> ### Author Response · Authors · 2020-11-18
> **Answer to reviewer #3**
>
> We would like to thank you for your time reviewing our work and will try to answer your concerns below.
>
> **Usage of the parameters in the model**
>
> Regarding the usage of the parameters, the different directions are not combined together, rather they are treated independently and their aggregation results are concatenated, as explained in equation (10) derived from the work of [1]. However, we understand that this point was not clear and we added a new figure in the introduction (Fig 1) which shows step-by-step how the different directions are generated and used within the GNN. We hope that this figure clarifies your concerns and helps any future reader avoid this confusion.
>
> **Degenerate eigenspace**
>
> Your concerns regarding the degenerate eigenspace are mostly addressed by our current method thanks to 2 factors, namely the row-wise normalization and the absolute value that we use. First, the row-wise normalisation of the matrices $B_{dx}$ and $B_{av}$ means that the amplitude of the vector does not affect the result. Second, as you mentioned, using the absolute value in the creation of $B_{av}$ and after the aggregation of $B_{dx}$ removes the sign ambiguity.
>
> However, your concern remains true in the case of eigenvalue multiplicity greater than 1 where a re-ordering of the nodes can change the resulting eigenvectors. To address it, we created *appendix B.4*. First, we argue that this multiplicity is very rare, with empirical evidence that between ZINC and PATTERN there is only 1 graph out of 26k with multiplicity of the second eigenvector greater than 1 and no graph with Fiedler vector with repeated multiplicity. For the CIFAR10 dataset we used coordinates, already provided to the nodes as features, to define an equivariant vector field.
>
> We further argue that in the case of multiplicity greater than 1, a good strategy would be to randomly shuffle (during training time) different linear combinations of all concerned eigenvectors. This shuffling will act as a data augmentation of all the possible eigenvectors in the eigenspace. Although too much randomness can make it difficult to train a network, the very rare occurrence of multiplicity >1 for low frequencies means that the network will mostly train with deterministic eigenvectors. Thus, we hope to answer your concern showing that the vast majority of the training will be done with an equivariant aggregations and the very rare cases can be augmented.
>
> **Weisfeiler-Lehman test**
>
> Regarding the evaluation of the expressivity with the Weisfeiler-Lehman test, we followed your suggestion and added the section 2.7 and the theorem 2.8 which prove that our method is more powerful than the 1-WL test, thus it is more expressive than most GNN models in regards to the WL test. Appendix C.8 provides an example of graphs that can be distinguished with our method using the first eigenvectors, with either $B_{dx}^1$ or $B_{av}^1$, but not with the mean, sum or max aggregations.
>
> We hope to have answered your concerns, please let us know if you have any remaining questions or concerns regarding our work
>
> [1] Corso, Gabriele, et al. "Principal neighbourhood aggregation for graph nets." Advances in Neural Information Processing Systems 33 (2020).

---

### Official Review · AnonReviewer4 · 2020-10-30
**The algorithm is not clearly described and may not be scalable.**

**Rating:** 5
**Confidence:** 2

**Review:**

This paper provides a theoretical framework that allows to directional convolutional kernels in any graph, e.g., generalize CNNs on an n-dimensional grid. In the framework, gradients of the eigenvectors of the graph Laplacian are used to define “directions” on the graph.

The proposed method is well-motivated and seems to be theoretically justified (I did not fully understand the details and check the proofs). My main concerns are:

1.	The theoretical development is difficult to follow. The proposed method is not clearly described. It is hard for readers to find in the paper what are the steps of the proposed algorithm and understand how it works. It would be better to describe the algorithm step by step.

2.	I have some doubts about the practical value of the proposed method because it requires eigen-decomposition of the Laplacian matrices. Though the authors provide complexity analysis in the appendices, it would be more informative to provide a runtime comparison with SOTA methods such as the vanilla GCN.

================ Post Rebuttal =============================================================

Thank the authors for the updates.

In the latest version, the algorithm flow is clearly stated in Figure 1, and now I can understand how the algorithm works. The authors also reported additional results on running time in the latest version, which are informative.

Here is what I think after reading the paper again.

1.	This paper proposes a novel idea. Defining directions on graphs is not a well-addressed problem in current GNN models, and using the gradients of the low-frequency eigenvectors of the Laplacian to define directions seems novel and interesting to me.

2.	The insight and analysis are not clear. Section 2.4 is still difficult to follow after the updates. More importantly, I am not sure about the correctness of the theorems and corollaries.

       The K-walk distance is supposed to reflect the difficulty of passing information between two nodes, and a larger distance means more difficulty. In the paper the K-walk distance is defined as the average number of times that a K-step random walk from one node to hit another (formal definition given in Page 18), which really puzzles me, because frequent visits indicate ease of message passing. Did the authors confuse hitting probabilities with hitting times?

---

> ### Author Response · Authors · 2020-11-18
> **Answer to reviewer #4**
>
> We would like to thank you for your time reviewing our work and will try to answer your concerns below.
>
> **Theoretical development**
>
> First, we are very sorry that you found the development difficult to follow. To answer your concern and make sure our work can be understood by a quick glance at the paper, we added a new image in the introduction (Fig 1). In Fig 1, we explain step-by-step in details, equations and illustrations the various parts of the methodology and architecture. The pre-computed steps are shown in blue and the GNN steps are shown in green, which better highlights how the implementation is done. We hope it helps clarify the paper.
>
> **Pre-computed running time**
>
> Regarding your second concern on the computation time and complexity, we added a new appendix “B3 Running time”.  In fact, the total computation time is very similar to standard GNNs since the decomposition of eigenvectors is done in a pre-computed step (see Fig 1), and it does not take too much time to compute O(kE). Also, we use the SciPy function *scipy.sparse.linalg.eigsh*, which is optimized to quickly compute the k-first eigenvectors of a real, sparse and symmetric matrix. The precomputation of the first four eigenvectors for all the graphs in the datasets takes 38s for ZINC, 96s for PATTERN and 120s for MolHIV using a single CPU core (Intel Xeon CPU @ 2.20GHz). Hence, the pre-computation is very fast considering it can take hours to train a model.
>
> **DGN running time**
>
> Regarding the computation time of the DGN itself, the time complexity is O(kE + kN) compared to O(E + N) for the GCN, since they use the same architecture but with more aggregators (see Fig 1, last 2 columns). However, in all our experiments and empirical results, we use the same parameter budget for all methods and we parallelize the aggregation, meaning that the computation time is roughly similar. With an Nvidia Tesla T4 GPU, we recorded the running time on all benchmarks and found that our DGN is on average 16% slower per epoch, but 8% faster to train due to faster convergence (see Table 1 in appendix B.3). Yet, even with a similar computation time and parameter budget, our method significantly outperformed the state-of-the-art in the standard benchmarks.
>
> We hope to have answered your concerns since we improved the clarity of the paper with Figure 1 and demonstrated that there is no scalability problem. Please let us know if you have any remaining questions or concerns regarding our work. Also, let us know whether Fig 1 is helpful to you and if we can improve it.

---

### Official Review · AnonReviewer2 · 2020-10-31
**Review for Directional Graph Networks**

**Rating:** 4
**Confidence:** 5

**Review:**

This work considers the limitation of graph neural networks that cannot consider the directions. It proposed directional graph networks to overcome this limitation. The concerns for this work are as below:

1.	I do not quite understand the motivation of this work. In section 2.1, the authors explain that a big limitation of current GNN methods is that they cannot process directions. This is not case. In most message passing GNNs, they can rely on the adjacency matrix during message passing process. If a specific direction is desired, it is feasible to modify or design the adjacency matrix accordingly. This fact is also true on the grid graphs. Thus, I don’t quite buy the motivation of this work. The authors need to clarify this motivation and illustrate why the direction limitation is not resolvable by manipulating the adjacency matrix.
2.	The data augmentation is proposed in Section 2.7. The authors claimed that the advantage of the proposed method is that it does not influence the data but applied on kernels. However, there may be some issues here for changing kernels. In this work, the kernel is acting as a kind of message passing directions on the graph. The message passing patterns are an important part of the graph. If the patterns or connections are changed, the graph will change consequently. From this perspective, if the proposed method changes the kernel and modify graph connections, this cannot be considered as a strict data augmentation since the modified graphs can have totally different properties as the original ones. Thus, I would like the authors to provide a clarification on this point.
3.	The number of datasets used in the experimental parts is quite limited. These datasets are not commonly used in the community. The authors may want to clarify this and add more datasets for comprehensive evaluations.

---

> ### Author Response · Authors · 2020-11-18
> **Answer to reviewer #2**
>
> We would like to thank you for your time reviewing our work and will try to answer your concerns below.
>
> 1. The motivation behind our work lies in the lack of structural information given to the aggregation process, and the ineffective way that MPNNs [8] treat directions as edge features. The novelty of our work lies in the proposal of strong structural inductive biases in the GNN framework dependent on the global topology of the graph. As observed in the added Figure 1, the $B_{dx}$ and $B_{av}$ are (as you mentioned) weighted adjacency matrices with possible negative weights. Producing such directional matrices is very novel, with our **novel** contributions regarding directional matrices below
>     - We developed a method to get a directional field with any undirected graph by using the gradient of the eigenvectors (See Fig 1, Fig 2). Hence, any undirected graph can have a set of directional fields *without* any prior knowledge of the graph or the task.
>     - These gradients were shown to generalize convolutional neural networks on grid graphs, *without* explicitly adding an inductive bias about the horizontal/vertical directions of the grids, and are the first generalization of CNNs in graph data that does not use inductive biases about the task.
>     - We developed the directional smoothing and directional derivative kernels, which are both interpretable and can be used separately. Again, this approach is novel and gives a new methodology of modifying the adjacency matrix to get directional aggregators with specific properties. We also provided a set of other directional aggregations in Appendix A, which are all unique in the literature.
>     - Other methods of anisotropic aggregation rely on the node features, not the global graph structure, such as the GAT [6] which uses attention mechanism or DropEdge [7] which randomly drops some edges.
>     - The proposed aggregation matrices, which use the eigenvectors for directions, are strictly more powerful than standard aggregations in terms of the 1-WL test (see Theorem 2.8).
>
> 2. We do not believe that our rotation/distortion of the kernel has a strong impact on the patterns and properties of the graph for multiple reasons, ranging from how we apply the augmentation, to the small and continuous regime in which augmentation is done.
>     - Having shown that DGNs are a generalization of CNNs and knowing the success augmentation has had with CNNs in Computer Vision, we believe it is reasonable to think that our augmentation approach is well motivated.
>     - As shown in Fig 1, we still use the regular adjacency matrix in the aggregation, which is not affected by the rotation, so the network can always understand the correct node connectivity.
>     - Definition 5 rotates a pair of vector fields simultaneously in the same plane, so the information that is lost to $F_1^\theta$ will be captured by $F_2^\theta$
>     - Data augmentation is always done in a small angle regime, e.g. in computer vision, images similar to CIFAR10 get rotated only by a maximum of 30 degrees [9, 10] with small amount of noise, so the general image does not change. This is again the case here, with Fig 7 showing that the rotation angle should ideally stay below 10 degrees, and the distortion below 20%, although bigger models should work with stronger augmentations. In these small regimes, the general patterns and properties of the graph remain similar.
>
> 3. We added the MolHIV dataset (Fig 6) from the popular Open Graph Benchmark [2] to the original 3 datasets from another popular GNN benchmarking paper [1]. Both papers are well known within the community, they have received multiple citations, they regroup different SOTA models with public leaderboards, and some recent GNN papers at NeurIPS2020 use them for benchmarking [3,4,5]. Both papers aim to become the new standard for benchmarking GNNs and are moving away from the traditional datasets of Cora, CiteSeer and PubChem since, according to [2], they are easily solvable by simple GNNs and have data quality issues. We hope that having 4 datasets from both [1] and [2] can convince you of the empirical evidence for our work, considering also the theoretical evidence.
>
> We hope to have answered your concerns, please let us know if you have any remaining questions or concerns regarding our work.
>
> [1] Dwivedi, Vijay Prakash, et al. "Benchmarking graph..." 2020
>
> [2] Hu, Weihua, et al. "Open graph..." NeurIPS2020
>
> [3] Corso, Gabriele, et al. "Principal neighbourhood..." NeurIPS2020
>
> [4] Li, Pan, et al. "Distance Encoding..." NeurIPS2020
>
> [5] You, Jiaxuan, et al. "Design space..." NeurIPS2020
>
> [6] Veličković, Petar, et al. "Graph attention networks." 2017
>
> [7] Rong, Yu, et al. "Dropedge: Towards deep..." ICLR2019
>
> [8] Gilmer, Justin, et al. "Neural message passing..." 2017
>
> [9] Shorten, Connor, and Taghi M. Khoshgoftaar. ‘A Survey on Image...’, 2019
>
> [10] O’Gara, Sarah, and Kevin McGuinness. ‘Comparing Data Augmentation...’, 2019

---

### Author Response · Authors · 2020-11-18
**Summary of the changes during the rebuttal**

Here we present a summary of the main changes brought to the paper during the rebuttal phase. More details are given in the specific answers to each reviewer.

### Introduction and Theoretical development

**Figure 1** is added, which helps to understand the methodology and architecture of the model. Most of the paper is summarized in this image, which includes a visualization of the eigenvectors, directional matrices and GNN architecture, as well as the important equations for each step.

**Section 2.7** (Comparison with Weisfeiler-Lehman (WL) test), along with the appendix C.8 are added, proving that the proposed DGN is theoretically more powerful than the 1-WL test, and consequently more discriminative than most competing GNNs.

**Definition 5** in section 2.8 is improved with a better notation, and by adding the row normalization to the previous version of the equations.

### Results and discussion

**Figure 5** added 2 rows to compare the effect of positional encoding versus directional aggregation. We observe that the directional message passing using eigenvectors is empirically much more powerful than the positional encoding using the same eigenvectors.

**Figures 5 and 6** added a new column with the MolHIV dataset from [1], and both of them show state-of-the-art results for the proposed DGN.

**Figure 7** added experimental results for the data augmentation using a distortion of the directional matrices (proposed in section 2.8). We observe that the overfitting of the derivative kernel is significantly reduced when applying a distortion of less than 20%.

### Appendix

**Section B.1** added a paragraph about the MolHIV dataset.

**Section B.3** (Running time) is added to compare the epoch and total time of different models. We observe only a 16% slowdown in epoch computation time for DGNs, but a faster convergence of the models leading to an 8% lower training time.

**Section B.4** (Eigenvector multiplicity) is added to explain how we deal with eigenvector multiplicities, and how often they occur.

**Section C.8** is added as a proof for the WL-test presented in section 2.7.

[1] Hu, Weihua, et al. "Open graph benchmark: Datasets for machine learning on graphs." Advances in Neural Information Processing Systems 33 (2020).

---

### Decision · Program_Chairs · 2021-01-07
**Final Decision**

**Decision:**

Reject

**Comment:**

The main merit of the paper is to try to address some important issues about GNN, e.g. expressivity power and data augmentation, from a novel perspective and using well grounded mathematical tools. Unfortunately, however, this novel perspective is also introducing some confusion about its meaning in the context of graphs. In fact, it is not clear how, in the general case, direction as introduced in the paper makes sense, especially when considering the data augmentation approach. Moreover, although well grounded mathematical tools are used, proofs of theorems, as well as justification of related corollaries, are not sufficiently clear to guarantee their correctness.

In summary, a potentially interesting contribution that needs more work to better clarify motivations, grounding to common graph concepts, better presentation of the theoretical results.